# Machine learning of genomic features in organotropic metastases stratifies progression risk of primary tumors

Biaobin Jiang [1,16], Quanhua Mu[1], Fufang Qiu[2], Xuefeng Li[3,4,5], Weiqi Xu[6,7], Jun Yu [8,9,10,11,12], Weilun Fu[13], Yong Cao[13] & Jiguang Wang [1,2,14,15 ✉]

Metastatic cancer is associated with poor patient prognosis but its spatiotemporal behavior remains unpredictable at early stage. Here we develop MetaNet, a computational framework that integrates clinical and sequencing data from 32,176 primary and metastatic cancer cases, to assess metastatic risks of primary tumors. MetaNet achieves high accuracy in distinguishing the metastasis from the primary in breast and prostate cancers. From the prediction, we identify Metastasis-Featuring Primary (MFP) tumors, a subset of primary tumors with genomic features enriched in metastasis and demonstrate their higher metastatic risk and shorter disease-free survival. In addition, we identify genomic alterations associated with organ-specific metastases and employ them to stratify patients into various risk groups with propensities toward different metastatic organs. This organotropic stratification method achieves better prognostic value than the standard histological grading system in prostate cancer, especially in the identification of Bone-MFP and Liver-MFP subtypes, with potential in informing organ-specific examinations in follow-ups.

[1] Department of Chemical and Biological Engineering, The Hong Kong University of Science and Technology, Hong Kong SAR, China. [2] Division of Life Science, The Hong Kong University of Science and Technology, Hong Kong SAR, China. [3] The Sixth Affiliated Hospital of Guangzhou Medical University, Qingyuan People's Hospital, 511518 Qingyuan, China. [4] State Key Laboratory of Respiratory Disease, Sino-French Hoffmann Institute, School of Basic Medical Sciences, Guangzhou Medical University, 511436 Guangzhou, China. [5] Department of Radiation Oncology, The University of Texas MD Anderson Cancer Center, Houston, TX 77030, USA. [6] Department of Hepatic Surgery, Fudan University Shanghai Cancer Center, 200032 Shanghai, China. [7] Department of Oncology, Shanghai Medical College, Fudan University, 200032 Shanghai, China. [8] Institute of Digestive Disease, The Chinese University of Hong Kong, Hong Kong SAR, China. [9] Department of Medicine and Therapeutics, The Chinese University of Hong Kong, Hong Kong SAR, China. [10] State Key Laboratory of Digestive Disease, The Chinese University of Hong Kong, Hong Kong SAR, China. [11] Li Ka Shing Institute of Health Sciences, The Chinese University of Hong Kong, Hong Kong SAR, China. [12] CUHK Shenzhen Research Institute, Shenzhen, China. [13] Department of Neurosurgery, Beijing Tiantan Hospital, Capital Medical University, 100070 Beijing, China. [14] State Key Laboratory of Molecular Neuroscience, The Hong Kong University of Science and Technology, Hong Kong SAR, China. [15] Hong Kong Center for Neurodegenerative Diseases, Hong Kong Science Park, Hong Kong SAR, China. [16]Present address: Tencent AI Lab, Shenzhen, Guangdong, China. ✉email: jgwang@ust.hk

Metastasis, the dissemination of tumor cells to distant organs, is attributed to the majority of cancer-related deaths[1]. This is in part due to late diagnosis of metastasis when the dissemination is out of clinical control. Early diagnosis of metastasis remains challenging via the current standard TNM system that grades patients according to the primary tumor size (T), lymph node spread (N), and detection of overt metastasis (M). One reason is that metastatic cancer might seed in distant organs much earlier than it becomes the overt metastasis at a clinically measurable size. Previous studies have observed that metastatic tumor cells can enter a dormant state without outgrowth once reaching the distant organs[2,3]. By far, given the standard TNM metrics, few solutions are available to quantitatively assess metastatic risk (e.g., when and where to spread) of a primary tumor before overt metastasis.

High-throughput technologies enable the identification of molecular signatures predictive of cancer metastasis and progression. In 2003, Ramaswamy et al. identified a gene expression signature associated with metastasis from microarray data by comparing 12 metastatic and 64 primary samples, and validated that the patients with this signature in an independent cohort of 279 primary solid tumors were associated with metastasis and poor clinical outcome[4]. Clinical tumor DNA-sequencing methods, such as MSK-IMPACT[5] and FoundationONE[6], have demonstrated its clinical utility in guiding treatment selection in both primary and metastatic cancers[7,8]. Moreover, copy-number alteration burden, a common genomic feature of cancer cells, proved to be highly predictive of the relapse of prostate cancer[9]. We, therefore, hypothesize that genomic variation of primary tumors could be used as the indicators for metastatic risk assessment.

To reliably estimate the potential time of overt metastasis, the risk assessment model ideally learns underlying patterns of tumor progression and migration from longitudinal sequencing data before and after metastasis[10]. However, currently available genomic databases have only a small number of such paired samples, which are insufficient to sort out reliable prognostic biomarkers applicable in a larger cancer population. Alternatively, there are many large-scale clinical DNA-sequencing data of cross-sectional primary and metastatic tumor samples, which could mitigate the shortage of longitudinal data. For example, recent analyses of MSK-IMPACT data in breast[11] and colorectal[12] cancers uncovered significant prognostic biomarkers indicative of treatment response and patient survival. Through learning the genomic difference before and after metastasis from those unpaired samples, computational models can then automatically assess the metastatic risk of a primary tumor by seeking metastatic features in its genome. Therefore, the unpaired sample data, even derived from different patients, may still be valuable resources to characterize tumor behaviors during clonal evolution and cancer progression.

Epidemiological studies have discovered that depending on the tissue of origins and other factors, metastatic tumor cells have a preference to seeding at certain distant organs, known as organotropism[13,14]. For example, the epidemiological data from 2010 to 2015 have shown that approximately 80% of synchronous brain metastases originated from lung primaries[15]. Metastases from the same tissue but colonizing at different organs may result in different patient prognosis[16,17]. For example, hepatitis metastases commonly lead to the significantly worse clinical outcome than other metastases in most cancer types[18].

In this work, we aim to develop a Metastatic Network model (MetaNet) to assess metastatic risk and potential destination organs through collecting and analyzing a total of 32,176 pan-cancer DNA-sequencing samples. Using this big-data cohort, we identify genomic biomarkers associated with common and organotropic metastases and validate their utility in metastatic risk assessment at an early stage using a machine-learning model to sort out a distinguishing subtype, namely Metastasis-Featuring Primary, with shorter disease-free survival than Conventional Primary patients. From the biomarkers associated with brain metastasis of lung cancer, we discover a significant enrichment of PI3K-pathway aberration and verify using our pharmacogenomic database that targeting *MTOR* is highly effective in treating lung cancer brain metastasis. Using the organotropic biomarkers, we establish a computational model that stratifies patients of primary prostate cancer into subgroups with propensities of bone or liver metastases to inform organ-specific examinations in follow-ups. To facilitate the metastatic risk assessment and other organotropic biomarkers validation, we develop a web application of MetaNet which is available at https://wanglab.shinyapps.io/metanet.

## Results

**Spreading pattern of pan-cancer metastasis.** To comprehensively profile the spreading pattern of metastatic cancers and enhance statistical power to identify underlying prognostic genomic biomarkers, we integrated the clinical and genomic data of 32,176 primary and metastatic cancer samples from four studies (Supplementary Fig. 1a): 10,946 samples from MSK-IMPACT (MSK), 18,004 samples from Foundation Medicine Inc. (FMI), 500 metastatic samples from the University of Michigan (MET500) and 3,336 primary samples of lung, breast, colon, and prostate cancers from The Cancer Genome Atlas (TCGA). We grouped tissues of origin (Supplementary Data 1) and sampling locations (Supplementary Data 2) of these tumor samples into general anatomic organ sites, and rule out the minor and unexplicit tissues from downstream analysis (Supplementary Fig. 1b). Next, we constructed a spreading diagram of pan-cancer metastasis (Fig. 1a), originating from 16 distinct primary sites and migrating toward 8 metastatic sites. Except for lymph nodes as the locoregional metastatic sites, the top-ranking distant metastatic organs are liver, bone, lung, and brain, all of which were intensively studied in organ-specific metastasis[19].

To investigate common cancer spreading patterns, we constructed a primary cancer network (PCN, Fig. 1b) based on the similarity of fractional distribution at metastatic sites among the 16 primary cancers. Through network clustering analysis, we identified two clusters of primary cancers, within which the primary-cancer organs exhibiting similar preference of metastatic direction are from the same functional system. One cluster is liver-tropic, including breast cancer and all the cancers in the digestive system from the esophagus, stomach, gallbladder, pancreas, and colon. The other cluster is mainly lung-tropic, including skin, thyroid, liver, head and neck, and the cancers from the urinary system (kidney, bladder, and prostate). Similarly, we constructed a metastatic site network (MSN) by clustering the 8 metastatic sites based on their similarities of metastatic cancer types they receive (Fig. 1c), and discovered that the four common metastatic organs, liver, lung, bone, and brain, received distinct types of primary cancer.

To investigate what are the genetic factors mediating this complex spreading pattern and to explore the underlying clinical implications, we developed MetaNet, a computational framework to predict whether and where a tumor will spread based on its genomic profile and clinical data at the primary stage (Fig. 1d). In general, MetaNet consists of two models: Model 1 to predict whether a primary tumor will metastasize via learning the genomic difference between primary and metastatic tumors, and Model 2 to predict where a primary tumor will colonize via capturing the genomic features among organ-specific metastases (Fig. 1d).

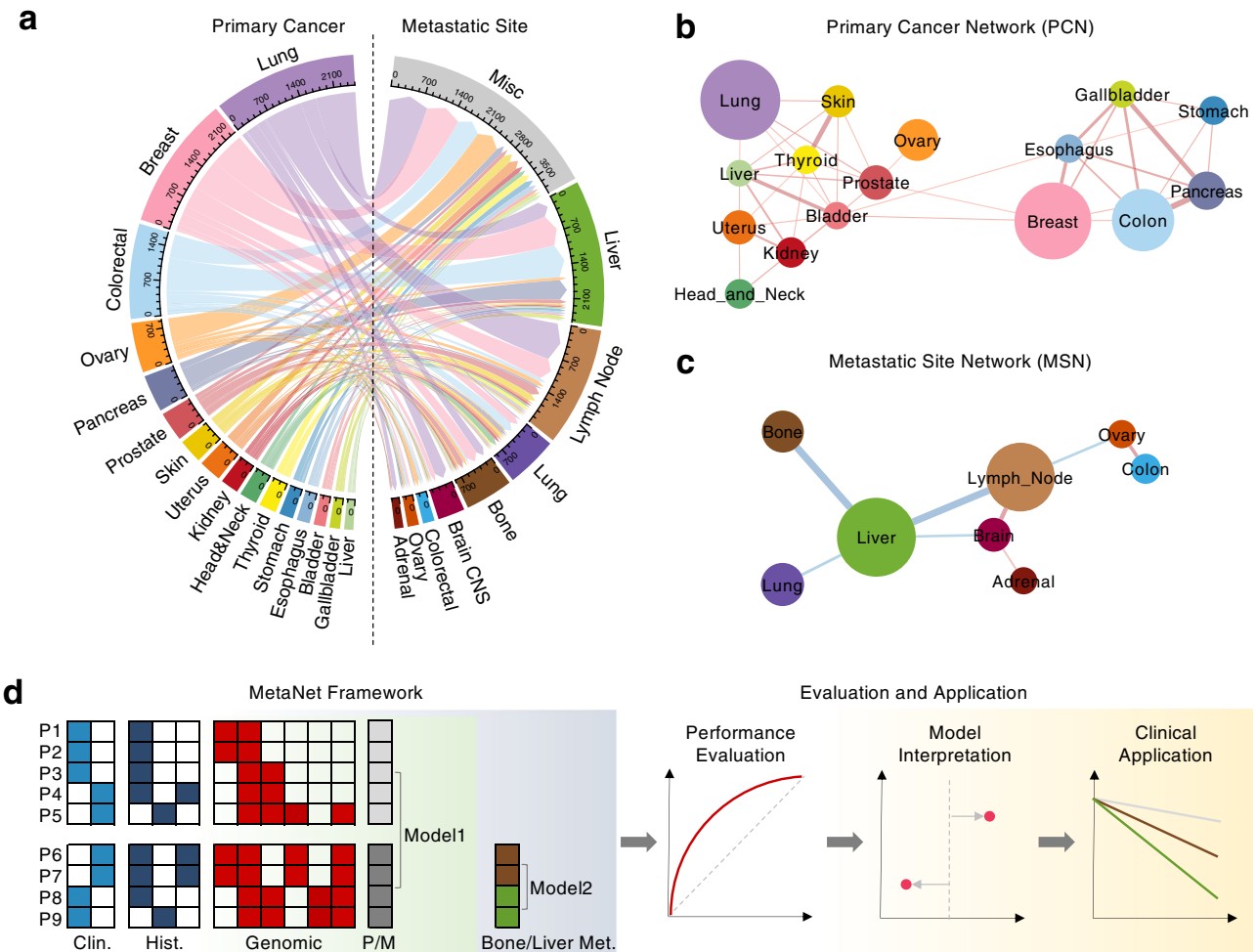

**Fig. 1 Spreading pattern of pan-cancer metastasis. a** Sankey diagram displays metastatic spreading directions from 16 primary cancer types toward 8 metastatic sites. Bandwidth is proportional to the number of metastatic tumor samples from one primary site to one metastatic site. Circle border thickness is proportional to the number of metastatic samples in that site. The color code representing the corresponding organ sites is used throughout the entire study. **b, c** Primary cancer network (PCN) represents the Pearson correlation coefficients (PCC) of fractional distribution of metastatic sites between each pair of primary cancers (**b**). Metastatic-site network (MSN) represents the PCC of fractional distribution of primary sites between each pair of metastatic sites (**c**). The correlations of *p*-value < 0.05 in PCN and *p*-value < 0.1 in MSN are shown. The significance is derived from a two-sided *t*-test without adjustment for multiple comparisons. The edge width is proportional to the absolute PCC. Red edge color denotes a positive correlation and blue color represents a negative correlation. Node size is proportional to the sample size of the primary tumor site (**b**) and metastatic site (**c**), respectively. **d** Schematic illustration of design, evaluation, and application of MetaNet. Each row of the grids represents the features of one patient consisting of the clinical (Clin.), histological (Hist.) and genomic features, together with the sample type: Primary (P) or Metastatic (M), and metastatic site, e.g., bone metastasis (Met.) or liver metastasis (Met.). Model 1 aims to learn the difference between primary and metastatic samples (green background), and Model 2 aims to learn the difference between different organotropic metastases (blue background). In the Evaluation and Application module (yellow background), Receiver operating characteristic (ROC) curve is used to evaluate the prediction performance. SHapley Additive exPlanations (SHAP) value is used to interpret the predictive contribution of each feature in each sample. And Kaplan–Meier plot (KM plot) is used to illustrate survival differences among different stratified groups.

Through scoring metastatic competence and organ-specificity of each tumor based on its genomic profile by the models, we further evaluate the prediction accuracy of the metastasis from the primary, and interrogate what are the associated genetic factors that contribute to the prediction. We finally validate our models through prognostic analysis using independent cohorts of primary tumors that are classified into different risk groups by MetaNet.

**Identification and characterization of metastasis-featuring primary tumors.** To identify genomic variants associated with metastasis, we compared the proportion of each mutation, copy-number alteration (CNA), chromosome-arm alteration, and the ten oncogenic pathway aberrations[20] in the 16 cancers in primary

and metastatic stages (Supplementary Fig. 2a). In general, there are more variants significantly enriched in the metastasis than those in the primary (Fig. 2a, b), indicating that metastasis evolving from primary cancer is a selective process along which the tumor gains metastatic competence through additional variations. Notably, the most significantly enriched variants in the metastasis include *ESR1* (estrogen receptor 1) mutation in metastatic breast cancer (FDR < 1e−6, *z*-test, Benjamini–Hochberg (BH) correction) and *AR* (androgen receptor) mutation and copy-number amplification in metastatic prostate cancer (FDR < 1e−6, *z*-test, BH correction). Previous studies reported that the *ESR1* mutations were commonly observed in recurrent breast cancer with resistance to hormonal therapy[21,22]. Similarly, *AR* variations have also proven to be the molecular mechanism of resistance to androgen-

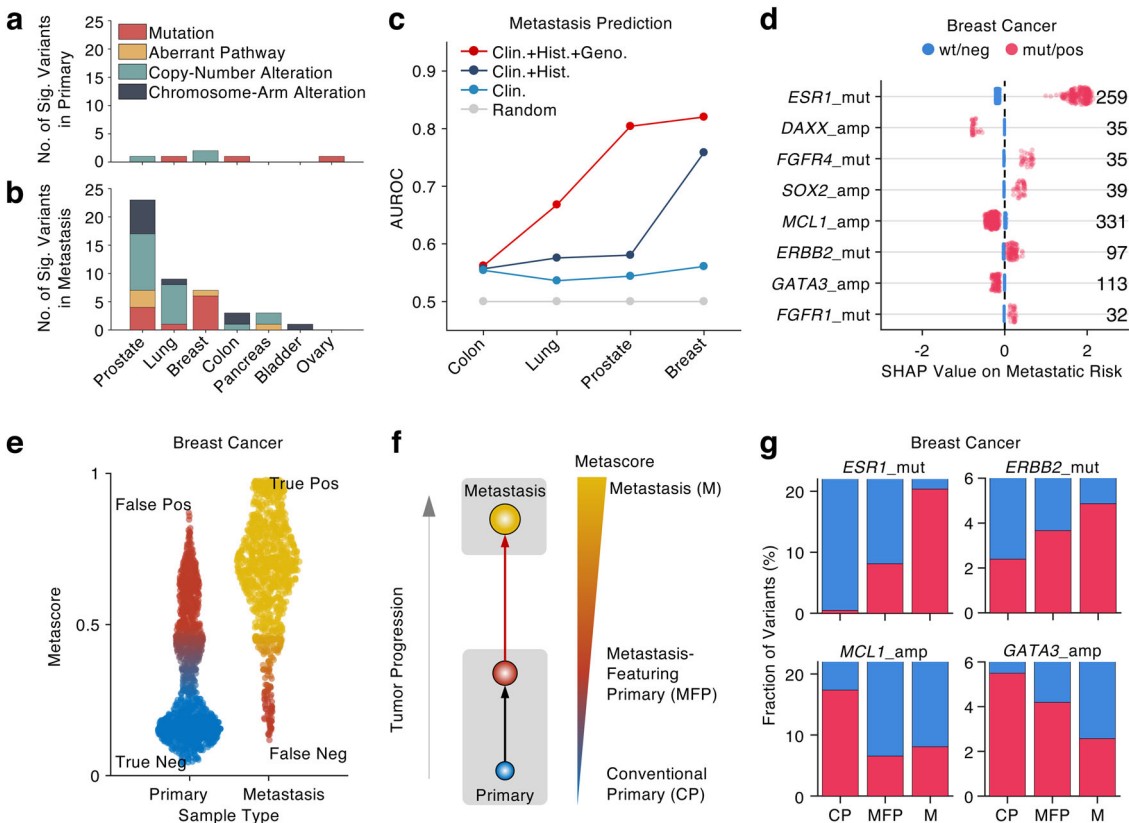

**Fig. 2 Identification and characterization of metastasis-featuring primary tumors. a, b** Number of significant (Sig.) variants enriched in primary (**a**) and metastatic (**b**) tumor samples. **c** Area under the ROC (AUROC) curves quantifies the performances of metastasis prediction from primary samples of breast, prostate, lung, and colon cancers in five-fold cross validation. **d** SHAP values represent selected predictive variants in primary versus metastasis prediction of breast cancer. Numbers on the right side denote the number of mutant or positive variants (red dots). **e** Metascore distributions in primary and metastatic tumor samples of breast cancer. Each dot represents the Metascore of one sample. **f** Schematic illustration of newly defined tumor category, termed metastasis-featuring primary (MFP) tumors, based on Metascore. **g** Fraction of selected variants in conventional primary (CP), metastasis-featuring primary (MFP), and metastasis (M) categories in breast cancer.

depletion therapy[23,24]. In addition, we also observed an increased number of copy-number and chromosome-arm alterations in the metastatic tumor genomes (Fig. 2b), which is consistent with a previous study reporting a highly unstable genomic structure in metastatic tumors[25]. Within those significant CNAs, we found that *MYC* amplification is the only variant across two different cancer types: metastatic prostate and pancreas cancers (Supplementary Fig. 2a), suggesting its common role in promoting cancer metastasis[26].

Given the observed differences in the genomic profile between primary and metastasis (Fig. 2a, b and Supplementary Fig 2a), we established MetaNet Model 1, a machine-learning module based on xgboost[27], a gradient boosting tree model to stratify patients with primary tumor into different metastatic risk groups using the tumor genomic profiles (Fig. 1d). In particular, we first trained the model to identify metastatic tumors from primary tumors in four common cancer types (breast, lung, colon, and prostate) using clinical, histological, and genomic features of the tumors. Learning the distinct features between primary and metastatic tumors, the model was then able to estimate the likelihood of one tumor being metastatic, termed Metascore. Using cross validation, we showed that compared to the baseline models trained only by the clinical and histological features without the genomic data, the genomics-based model can accurately identify the metastatic tumors from the primary in the breast and prostate cancers (Area Under the Receiver Operating Characteristic curve, AUROC > 0.8), rather than in

the lung and colon cancers (Fig. 2c and Supplementary Fig. 2b). This result demonstrated that the primary and metastatic breast and prostate tumors are genomically different, while in lung and colon cancer the genomes are alike, which is similar to our observation in the comparison of the genomic profiles (Fig. 2b). From an evolutionary perspective, it suggests that unlike lung and colon cancers, breast and prostate cancers may follow certain evolutionary modes in which only novel clones resistant to hormone treatments can thrive in the metastasis. In terms of clinical implication, disease-free survivals of breast and prostate cancer patients are generally longer than those with lung and colon cancers[28], during which the metastases of breast and prostate cancers have longer time to evolve and acquire more variants than those of lung and colon cancers under the assumption of constant mutation rate.

To further understand what genomic variants are used in the model for metastatic risk prediction, we used SHapley Additive exPlanations (SHAP) value[29] to untangle the tree-based model by visualizing gene-wise contribution to the metastatic risk of breast cancer (Fig. 2d). A positive SHAP value indicates that the genomic feature has a positive contribution to the metastatic risk, while a negative value represents a negative impact on the risk. Consistent with our genomic comparison between primary and metastasis (Supplementary Fig. 2a), the *ESR1* mutation is found by the mean SHAP value as the most predictive feature of metastatic breast cancer, followed by *FGFR4* mutation, *SOX2* amplification, *ERBB2* mutation, and *FGFR1* mutation (Fig. 2d).

*ERBB2* mutation has been found to be associated with resistance to hormone therapy through a distinct mechanism from the *ESR1* mutation[30]. In contrast, the top predictive variants of low metastatic risk in breast cancer are *DAXX*, *MCL*, and *GATA3* amplification. Interestingly, a previous study has uncovered that *GATA3* plays a suppressive role in breast cancer metastasis by inducing microRNA-29b expression which targets a set of pro-metastatic regulators[31].

Even though the genomics-based model achieved AUROC of 0.82 and 0.8 in distinguishing metastatic versus primary breast cancers and prostate cancer, respectively (Supplementary Fig. 2b1–2), the misclassification rate is not ignorable. Examining the Metascore distributions in the true primary and metastatic sample categories, we showed that the misclassification rate is mainly contributed by a high false-positive rate in breast cancer (Fig. 2e) and prostate cancer (Supplementary Fig. 2c), suggesting that the model overrates a subset of primary tumors as metastatic ones. This overrated subset of primary tumors, even though labeled as primary, might carry metastasis-enriched features, which makes them genomically more similar to the metastatic tumors other than the primary tumors. We, therefore, deemed this subset of primary tumors as Metastasis-Featuring Primary (MFP) tumors ($n = 382$, Fig. 2f) and the other primary as Conventional Primary (CP) tumors ($n = 1,255$, Fig. 2f) based on a Metascore cutoff of 0.5 (Fig. 2e). To illustrate whether the MFP tumors in fact carry the metastasis-enriched features, we calculated the fraction of the top predictive variants (Fig. 2d) in the MFP, CP, and metastatic (M) breast cancers. Consistently, we found that the MFP tumors harbor more metastasis-enriched features, such as the *ESR1* and *ERBB2* mutations than the CP tumors (Fig. 2g). Conversely, the MFP tumors carry less primary-enriched features, such as *MCL1* and *GATA3* amplifications than the CP tumors (Fig. 2g), which indicates that more metastasis-enriched features and less primary-enriched features together shift the MFP tumors away from the conventional primary toward real metastasis on the genomic scale defined by our Metascore (Fig. 2f).

**Transcriptomic characteristics and prognostic value of metastasis-featuring primary tumors**. To explore the biological and clinical implications of the MFP tumors, we collected the genomic, transcriptomic, and clinical data of TCGA breast cancer cohort[32,33] consisting of 1,079 primary breast cancer samples. Feeding the identical features from the clinical, histological, and genomic data of TCGA samples into the trained model, we estimated the metastatic risk of each TCGA sample by computing their Metascore. The top predictive genomic features identified in the training phase (Fig. 2d) contributed similar predictive power to the metastatic risk estimation of each TCGA sample (Fig. 3a), highlighting the robustness of these predictive features regardless of the variation caused by batch effect and other covariates. In particular, the seven TCGA primary breast cancer samples carrying *ESR1* mutation are all deemed as MFP tumors, highlighting *ESR1* mutation is a remarkable feature that can provide early warning signal of high metastatic risk in the patients with primary tumors, but their metastatic lesions are not detectable yet. Indeed, *ESR1* mutation has been used to monitor the resistance of hormone treatment via liquid biopsy, the measurement of cell-free DNA in the blood of cancer patients[34].

To understand the functional consequence of primary tumors harboring metastatic features we subsequently studied gene expression data. As different receptor-defined subtypes of breast cancers exhibit distinct expression patterns, we split TCGA breast cancer samples into the four classical subtypes[32,33]: luminal A, luminal B, HER2-enriched, and basal-like, and then compared the transcriptomic profiles between the MFP and the CP tumors within each subtype. Notably, we found that the upregulated genes in the MFP tumors are significantly enriched in the epithelial–mesenchymal transition (EMT) in both HER2-enriched and basal-like subtypes, while the downregulated genes are significantly enriched in the functions related to cell-cycle proliferation, such as G2M checkpoint and E2F targets (FDR < 0.0001, Gene Set Enrichment Analysis (GSEA), Fig. 3b, c). This pattern was not found in the GSEA of the other two hormone-related subtypes and breast cancer in general (Supplementary Fig. 3a–c). A previous study observed the same reverse pattern between EMT and cell proliferation through modulating *CDH1* expression in MDA-MB-468, a triple-negative breast cancer cell line[35], which in part supports our observation.

To validate whether the MFP tumors have a high risk of metastasis, we compared the clinical outcomes of the patients classified into the MFP and the CP groups. Filtering the samples with the survival data available in TCGA, we showed that the patients with MFP tumors have significantly shorter disease-free survival (DFS) than those with CP tumors ($p$-value < 0.0001, log-rank test, Fig. 3d). Similarly, worse clinical outcomes with shorter DFS were found in the patients with the MFP prostate (Supplementary Fig. 3d) and the MFP lung cancers (Supplementary Fig. 3e), which collectively demonstrates that the MFP tumors are more progressive than the CP tumors. To validate that the Metascore, our genomic estimation of metastatic risk, is an independent predictor of disease progression, we compared the DFS between the patients with MFP tumors and those with CP tumors within each breast cancer subtype and found consistently worse DFS in the MFP group within each of the four subtypes (Fig. 3e and Supplementary Fig. 3f). Moreover, we used multivariate Cox regression to collectively evaluate the predictive power of the breast cancer subtypes and the Metascore-defined MFP/CP stratification. Strikingly, the hazard ratio of MFP over CP is 3.9 (2.2–7.0, 95% confidence interval), which is significantly higher than the base value of 1, and is independent of the subtypes (Fig. 3f). Even though the previous study has demonstrated significantly worse overall survival of basal-like breast cancer than those of hormone-positive subtypes[36], the subtypes, however, did not exhibit strong predictive power to the disease progression when standing with our genomics-based stratification (Fig. 3f). Taken together, we demonstrated that our genomic stratification of metastatic risk is significantly powerful and independent of conventional hormone-based subtyping in breast cancer.

**Profile of metastatic organotropism**. To explore the spreading preference of cancer metastasis, we curated two large-scale and independent metastatic cancer data sets from MSK ($n = 2,919$) and FMI ($n = 4,100$) cohorts, in order to investigate whether this organ-specific metastasis, namely metastatic organotropism, is a statistically robust phenotype. Comparing the fractional differences of metastatic cancers located in the 8 metastatic sites from the 16 tissue of origins between the two independent cohorts (Supplementary Fig. 4a), we discovered a remarkably significant correlation (Pearson Correlation Coefficient, PCC = 0.913, $p < 0.0001$, Fig. 4a) in a pan-cancer scale. The most correlated metastatic cancer type is the liver metastasis of pancreatic cancer (143 out of 180 in MSK versus 263 out of 329 in FMI, $p = 0.89$, proportion test). Collectively, given that 15 out of the 16 cancer types (except the prostate cancer) exhibit a significant correlation of metastatic site distributions between MSK and FMI cohorts (PCC > 0.5, $p < 0.05$, Supplementary Fig. 4a), we concluded from a big-data perspective that dissemination direction in the majority of metastatic cancers is strongly organotropic in a statistically

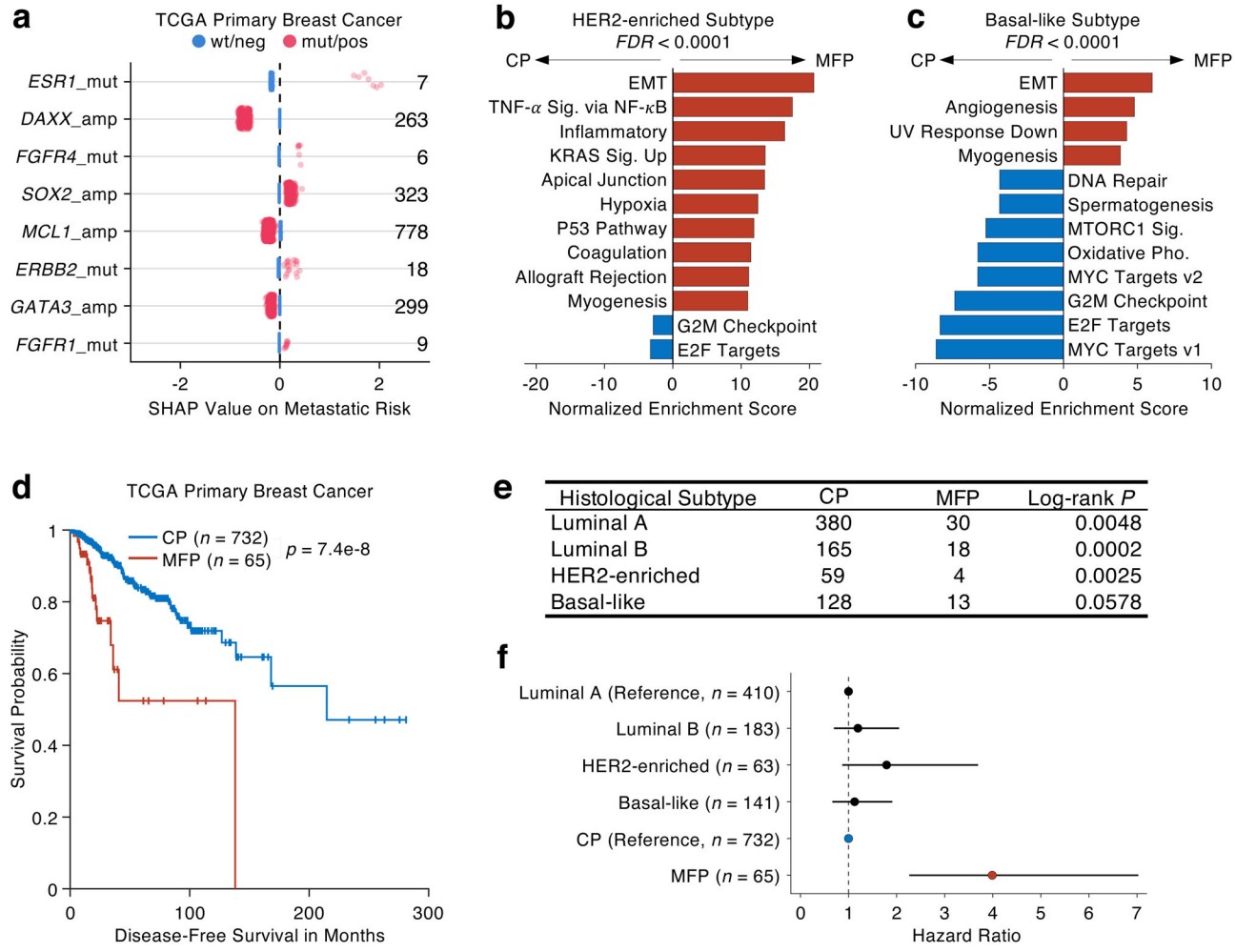

**Fig. 3 Transcriptional characteristics and prognostic value of metastasis-featuring primary tumors. a** SHAP values represent selected predictive variants in the TCGA breast cancer cohort. Numbers on the right side denote the number of mutant or positive variants (red dots). **b, c** Normalized enrichment scores of significant hallmark functions (FDR < 0.0001) in Gene Set Enrichment Analysis (GSEA) which compares MFP tumors versus CP tumors in HER2-enriched subtype (**b**) and basal-like subtype (**c**) of TCGA primary breast cancer cohort. Bars in red/blue represent activated hallmarks in MFP/CP tumors. **d** KM plot displays disease-free survival (DFS) difference between MFP tumors versus CP tumors of TCGA primary breast cancer. The censored data are denoted as + sign. The significance is derived from the two-sided log-rank test. **e** Table displays DFS differences between MFP tumors versus CP tumors in four different breast cancer subtypes of the TCGA cohort. **f** Hazard ratios and 95% confidence intervals derived from multivariate cox regression using breast cancer subtypes and genomics-based stratification: MFP versus CP. The sample sizes are denoted in the brackets.

robust manner, which implies that organotropic metastasis is highly non-random and in part driven by certain potential factors including tissue of origin, vascular pattern, genetic background, and congenial microenvironment[19,37,38].

To further visualize which organ is the predominant metastatic destination in each cancer type, we compared the fractions of cancer samples at the four common distant metastatic organs: bone, brain, liver, and lung (Supplementary Fig. 4b), by projecting the normalized fractions into a tetrahedron space (Fig. 4b). Interestingly, we uncovered two cancer groups: one is liver-tropic and the other is lung-tropic, which is consistent with the discovery in the primary cancer network (Fig. 1b). The liver-tropic group consists of five cancer types all from the digestive system (gallbladder, pancreas, stomach, colon, and esophagus), which is in part due to vascular structure and anatomic proximity. The lung-tropic group consists of the cancer types from head and neck, thyroid, uterus, skin, and kidney, most of which are located close to the lung. These two groups indeed explained that the cluster formation in the primary cancer network (Fig. 1b) is due to the predominant single-organ

tropisms in the liver and lung. In addition, we observed widely reported organotropisms, including bone metastasis of prostate cancer and brain metastasis of lung cancer[39,40]. The other cancer types consisting of bladder, ovary, and liver cancers, were not located close to any single corner, indicating their metastatic organotropisms are not dominated by one single organ (Supplementary Fig. 4b).

Given that metastatic organotropism is a stable biological phenomenon (Fig. 4a), we further investigate its underlying clinical value, i.e., whether the metastases at different organs impact patient survival. Using the overall survival (OS) data available in the MSK cohort, we compared the OS differences of the four common cancers spreading to the four common metastatic organs based on the metric of the area under the Kaplan–Meier plot (equivalently as mean survival) instead of median survival which cannot be computed in long-surviving cancers, such as prostate cancer. Generally, in all the four cancer types the patients with metastatic cancer have remarkably shorter survival than those with primary cancer (Fig. 4c and Supplementary Fig. 4c1–4). Particularly, the patients with metastatic

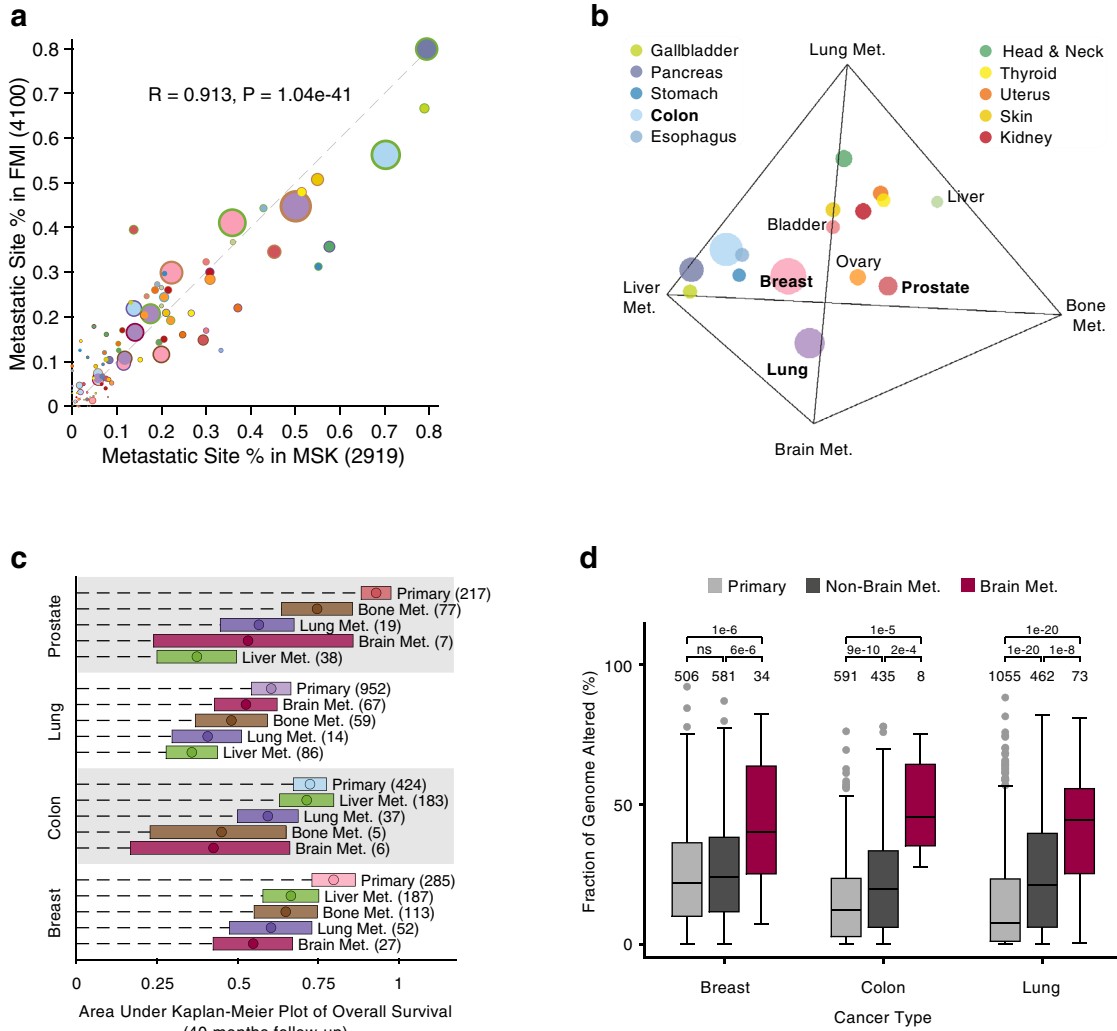

**Fig. 4 Profile of metastatic organotropism. a** Correlation of metastatic site fractions between MSK and FMI cohorts. Node size is proportional to the sample size of metastatic tumor. Node color denotes the primary tumor site and node border color denotes the metastatic site. The significance of PCC is derived from a two-sided *t*-test. Color code is the same as Fig. 1a. **b** Fractions of bone, brain, liver, and lung metastases of 16 primary cancers are projected into the tetrahedron space. Node size is proportional to the sample size of metastatic tumors. Color code is the same as Fig. 1a. **c** Mean (circle) and 95% confidence interval (rectangle) of overall survival of four cancer types in the primary site and four common metastatic (Met.) sites from MSK cohort given 40-month follow-up, equivalently as the area under the KM plot, is normalized by 40 months. **d** Fraction of genome altered of primary, non-brain metastases and brain metastasis in three common cancer types from MSK cohort. ns not significant. Significance is derived by a one-sided Wilcoxon rank-sum test. Boxes represent upper and lower quartiles; the lines inside denote median; whiskers correspond to 1.5 times the interquartile range.

prostate cancer at the liver have significantly worse survival than those at the bone (Fig. 4c), which implies that predicting potential metastatic sites can provide prognostic value for prostate cancer patients. Unexpectedly, brain metastasis does not always indicate worse survival than the other metastases: in breast and colon cancers the brain metastases have worse survival than the liver metastases, while this observation is opposite in metastatic lung cancer (Fig. 4c).

We next investigate whether genetic variation contributes to metastatic organotropism. Given the abundance of copy-number and chromosome-arm alterations significantly enriched in metastatic cancers (Fig. 2b), we further investigate whether those alterations are uniformly distributed in all metastatic sites or specifically enriched in a certain one. Using the fraction of genome altered (FGA) estimated in the MSK-IMPACT study[7], we found a dramatic increase of FGA in the brain metastases compared to the non-brain metastases and the primary tumors in 10 out of the 16 cancer types (Fig. 4d and Supplementary Fig. 4d),

especially in the lung, breast, and colon cancers (Fig. 4d), all of which are the top-ranking origins in brain metastasis (Supplementary Fig. 4e)[41]. The previous study has demonstrated that chromosomal instability, featured by high FGA, is a driver of metastasis through a cytosolic DNA response[25]. A recent study discovered *MYC* amplification, in particular, is required in the brain metastasis of lung cancer using patient-derived xenograft mouse models[42], which enlightened us to further characterize each organ-specific metastasis from a gene-wise perspective.

**Genomic characterization of metastatic organotropism.** To comprehensively identify variants associated with metastatic organotropism in our curated large-scale dataset, we selected the metastatic samples located at bone, brain, liver, and lung, and screened for the variants whose fractions in the four metastatic sites have a significant deviation from the average fraction in the metastases. Using false discovery rate control, we identified 93

organotropic variants and features in total with fractional bias in certain metastatic sites significantly deviating from the average (Chi-squared test, FDR < 0.1, variant fraction in the metastases >1%, Supplementary Fig. 5a and Supplementary Data 3), most of which are found in the metastatic cancers originating from the colon ($n = 19$), the breast ($n = 15$) and the lung ($n = 13$). Almost two-thirds of the organotropic variants (59 out of 93) have fractional enrichments in the brain metastasis, whereas only 6 variants are enriched in the lung metastasis and 5 in the liver metastasis (Supplementary Fig. 5a). Within those 59 brain-tropic variants, 15 are CNAs, suggesting those altered genes might be the key factors among the abundant CNAs enriched in the brain metastasis (Fig. 4d).

Among the 15 organotropic variants in breast cancer, 10 are most abundant in brain metastasis, 3 in liver metastasis, and 2 in bone metastasis (Supplementary Data 3). To visualize the organ-specific enrichment of each variant, we calculated the odds ratio of the variant fraction in one metastatic site over that not in the site, termed organotropic odds ratio (OGTOR), and projected the normalized OGTORs into a tetrahedron with the four corners representing the four metastatic sites (Supplementary Fig. 5b). None of the variants are shown to locate close to the lung metastasis corner, and the most abundant variants and features in the lung metastasis, such as p53, Ras, and Myc pathways (67%, 56%, and 20%), are found to be more abundant in the brain metastasis (80%, 73%, and 37%). Projecting the OGTORs into a triangular space (Fig. 5a) instead and highlighting the variants with variant fraction larger than 5% and FDR less than 0.05, we clearly showed that the top liver-tropic variant of the breast cancer is *ESR1* mutation, the bone-tropic variant is *CDH1* mutation, and the brain-tropic variants include *TP53* mutation, *CDK12,* and *ERBB2* amplifications. Particularly, the *ESR1* mutation dramatically shifts the distribution of metastatic breast cancer destinations with a sharp increase in the liver (from 52% to 75%) accompanied by fractional decreases in brain and lung ($p < 0.0001$, Chi-squared test, Fig. 5b). Further examining each *ESR1* mutation in the breast cancer samples ($n = 405$, 129 from MSK, 262 from FMI, and 14 from MET500), we found that the liver metastasis enrichment is in fact primarily contributed by four hotspot positions located at the ligand-binding domain (LBD): D538, Y537, L538 and E380 ($n > 20$, Fig. 5c), all of which are spatially close to each other and have been demonstrated to give rise to estrogen-independent activation of downstream signaling and promote cellular proliferation[43]. The *CDH1* mutation enriched in the bone metastasis, was identified as a featuring loss-of-function mutation in the invasive lobular carcinoma[33] that leads to dysregulation of cell-cell adhesion with a discohesive phenotype[44]. The brain-tropic variants *TP53* mutations and *ERBB2* amplification are in fact enriched in triple-negative and HER2-enriched breast cancers, respectively[32]. A previous epidemiological study showed that bone metastasis of breast cancer is common across all the subtypes except the basal-like one[45], whereas liver metastasis is enriched in hormone-positive subtypes[46] and brain metastasis is enriched in HER2-enriched[47] and triple-negative subtypes[48], all of which are highly consistent with our organotropic variation enrichment analysis.

Among the 13 organotropic variants in lung cancer, 10 are most abundant in brain metastasis and 3 in bone metastasis (Supplementary Data 3). No variant is significantly enriched in the liver metastasis, and we ruled out the lung metastasis due to its small sample numbers ($n = 32$) and missing annotation regarding regional relapse or distant metastasis from one site of the lung to the other. A significantly high mutation burden (larger than 20 mutations per megabase) was identified in the brain metastasis ($p < 0.0001$, Chi-squared test, Supplementary Fig. 5c), featured by the enriched *CREBBP* and *EPHA5* mutations (Fig. 5d).

The *STK11* mutation was significantly enriched in the bone metastasis ($p = 0.0002$, Chi-squared test, Supplementary Fig. 5d), and the aberration of its involved PI3K pathway was found to significantly increase the fraction of brain metastasis ($p = 0.0379$, Chi-squared test, Fig. 5e). Interestingly, the *STK11* mutation, even though more enriched in the bone metastasis, are found to be the most abundant variant in the brain metastatic samples of lung cancer harboring the aberration of PI3K pathway ($n = 146$, Supplementary Fig. 5e). The previous study has demonstrated that *STK11* is a tumor suppressor and its loss-of-function mutation is involved in the morphological change from adenocarcinoma to squamous cell carcinoma, which further promotes lung cancer metastasis[49]. Through analyzing our previous pharmacogenomic dataset that screened 60 anti-cancer drugs in 462 patient-derived cell lines (PDCs)[50], we showed that five PI3K-pathway inhibitors rank within the top seven most efficacious drugs for the 23 PDCs of lung cancer brain metastasis (LUBM). Three of the five PI3K-pathway inhibitors (gedatolisib, everolimus, and vistusertib) target *MTOR*[51], a downstream effector of the PI3K pathway. Further comparison of the drug efficacies in the 23 LUBM PDCs versus those in the other 439 PDCs showed that the three *MTOR* inhibitors exhibit significantly high specificity (FDR < 0.01, $t$-test, Fig. 5f and Supplementary Fig. 5f). Collectively, this result verified an enrichment of the aberrantly activated PI3K pathway and its clinical actionability in the brain metastasis of lung cancer.

Among the 19 organotropic variants in colon cancer, 11 are most abundant in brain metastasis, 7 in bone metastasis, and 1 in liver metastasis (Supplementary Data 3). No significant variants are found in the lung metastasis (Supplementary Fig. 6a). Besides the aberration of the TGF-β pathway enriched in the liver metastasis, featured by *SMAD4* mutation and deletion, the other significant variants are mainly amplifications located at chromosome 13q, together with Ras pathway activation featured by *KRAS* mutation (Fig. 5g). Among those brain-tropic amplifications, the most significant one is *CDK8* amplification which gives rise to a fractional increase of the brain metastasis from 2% to 11% ($p = 0.0004$, Chi-squared test, Fig. 5h). Even though *CDK8* is amplified together with its chromosomal neighbors *FLT1* and *FLT3*, we collected the transcriptomic data from TCGA[52] and showed that only the amplification of *CDK8*, rather than those of *FLT1* and *FLT3*, are functional through elevation of the corresponding expression (Supplementary Fig. 6b). The previous study has demonstrated the oncogenic role of *CDK8* amplification in colon cancer cell proliferation as a positive mediator of β-catenin-driven transformation in the WNT pathway[53]. Using paired samples of primary and brain metastasis colon cancer from the same patients in two recent studies[10,54], we demonstrated that the *CDK8* amplification is not a newly emerged event in the brain metastasis but inherited from the primary tumors (12 out of 14, Supplementary Fig. 6c). Comparing the transcriptomic profile of *CDK8*-amplification primary colon cancer in TCGA versus the non-amplified cases, we showed that *CDK8* amplification is in fact associated with the promotion of epithelial–mesenchymal transition (GSEA, FDR < 0.0001, Supplementary Fig. 6d) and down-regulation of cell proliferation (GSEA, FDR < 0.05, Supplementary Fig. 6e), implying a role of *CDK8* amplification in distant metastasis of colon cancer. Furthermore, we used the clinical data of TCGA to show that the colon cancer patients with *CDK8* amplification were diagnosed with more lymph node spread ($p = 0.006$, proportion test, Supplementary Fig. 6f) and have significantly shorter DFS ($p = 0.003$, log-rank test, Fig. 5i). Collectively, all of the evidences pinpointed that the colon cancers with *CDK8* amplification are more progressive with strong potential in distant migration toward the brain. Given that colon cancer metastasis follows a sequential cascade from the colon to

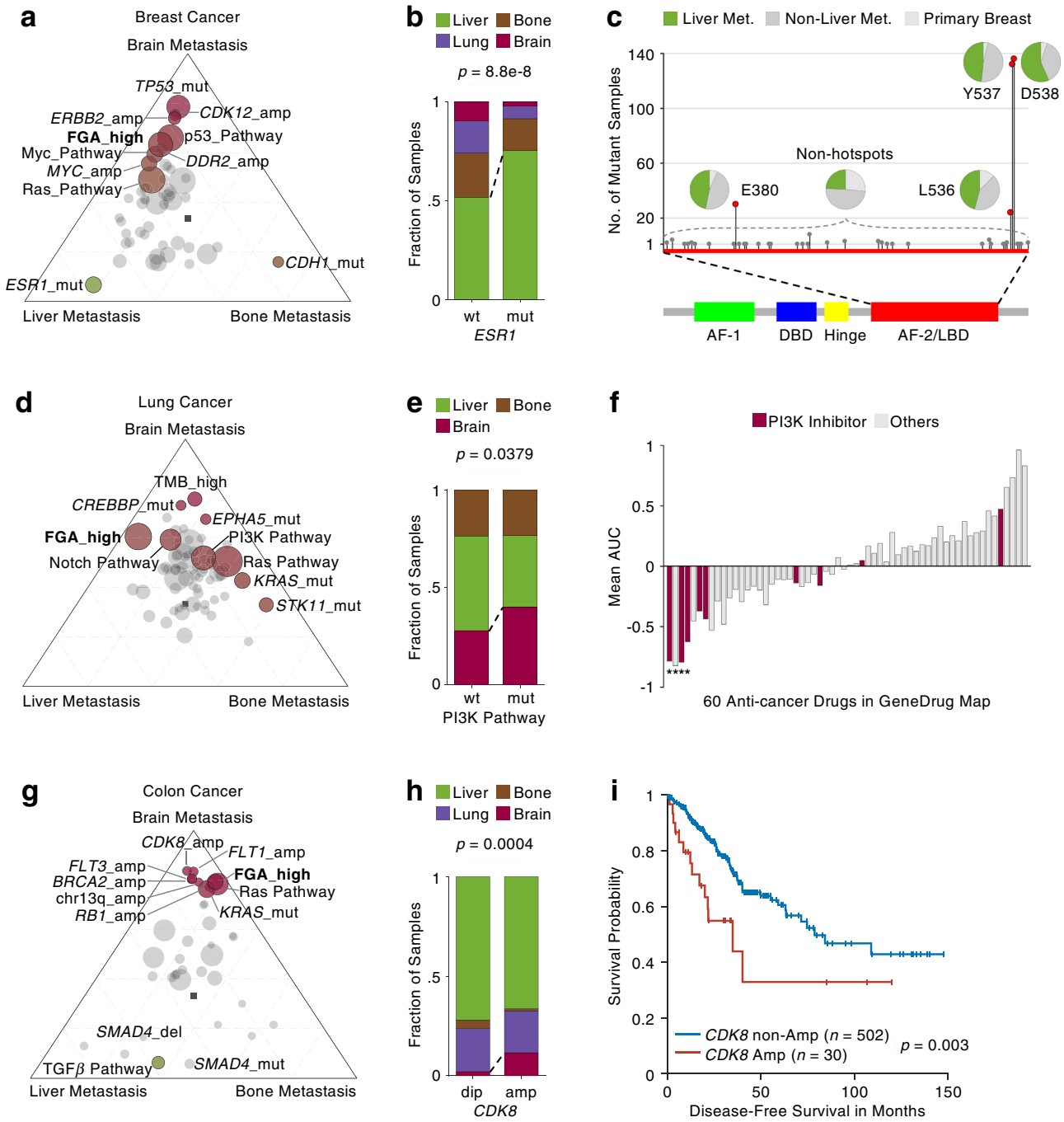

**Fig. 5 Genomic characterization of metastatic organotropism. a, d, g** Projection of odds ratio of mutant number in one metastatic site over that not in the site (OGTOR) for variants in brain, liver, and bone metastases of breast cancer (**a**), lung cancer (**d**), and colon cancer (**g**). **b, e, h** Fraction distribution of mutant and wildtype of *ESR1* in metastatic breast cancer samples (**b** wt: wildtype samples versus mut: mutant samples), PI3K pathway in metastatic lung cancer samples (**e** wt: wildtype samples versus mut: mutant samples) and *CDK8* amplification in metastatic colon cancer samples (**h** dip: diploid samples versus amp: amplified samples). All the significances are derived from two-sided Chi-squared test. **c** Illustration of *ESR1* mutations in a ligand-binding domain on the gene body. Pies represent the fractional distribution of primary, non-liver and liver metastatic samples with *ESR1* mutations on the corresponding hotspot/non-hotspots position. **f** Waterfall plot represents the mean area under the dose-response curve (AUC) which denotes the averaged efficacy of 60 anti-cancer drugs in 23 lung-cancer-brain-metastasis (LUBM) patient-derived cell lines (PDCs). Asterisk denotes significance in AUC comparison between 23 LUBM PDCs versus 439 other PDCs (*p* < 0.01, one-sided *t*-test, BH correction). The 60 drugs are ranked at an ascending order of one-sided *t*-test *p*-values. **i** KM plot represents the DFS difference between *CDK8* amplified versus non-amplified patients with primary colon cancer in the TCGA cohort. The censored data are denoted as + sign. The significance is derived from the two-sided log-rank test.

the liver, and lung[19], we inferred that brain metastasis of colon cancer also follows this cascade driven by the blood vasculature and keeps migrating from the lung into the heart and eventually into the brain through the neck artery. The *CDK8*-amplification fractions in the primary, liver, lung, and brain metastases exhibit a significant increased trend ($p = 0.003$, trend test, Supplementary Fig. 6g), which suggests that *CDK8* amplification is positively selected during the cascade.

**Organotropic stratification of primary tumors.** Even though we only identify one organotropic variant, the *MSH6* mutation in lung metastasis of prostate cancer (FDR = 0.07, Supplementary Data 3), we observed from the survival analysis (Fig. 4c) that the patients with primary prostate cancer, the bone metastasis, and the liver metastasis suffer a dramatic decrease of the mean survival time as 33, 24, and 13 months, respectively, given a 40-month follow-up (Figs. 4c and 6a). The pairwise comparisons between these three groups all yielded significant differences ($p < 0.001$, log-rank test, Fig. 6a), which is consistent with previous epidemiological statistics[17]. Enlightened by this fact, we, therefore, developed MetaNet Model 2 (Fig. 1d), an organotropism-based prognostic system that stratifies patients into different risk groups depending on the propensity of metastatic destination. Using an ordinal regression model framework, we trained the MetaNet Prognosis module using the combined dataset of the MSK and FMI prostate cancer cohorts (Fig. 6b), and achieved an accuracy of 64.3% in the three-class prediction task. Next, we applied our MetaNet prognosis module to an independent cohort of primary prostate patients from TCGA[55], and stratified them into three risk groups (Fig. 6b): conventional primary (CP, $n = 237$), bone metastasis-featuring primary (Bone-MFP, $n = 174$), and liver metastasis-featuring primary (Liver-MFP, $n = 83$). Strikingly, the three risk groups have a similar decreasing trend in DFS, and pairwise comparison of the DFS between the three groups yield significant differences: $p = 0.04$ in CP vs. Bone-MFP, $p = 9e-4$ in Bone-MFP vs. Liver-MFP, and $p = 6e-8$ in CP vs. Liver-MFP (Fig. 6c). This suggests that the organotropism stratification can inform clinicians to perform an organ-specific examination during the follow-ups of high-risk patients.

To display the mechanism of our genomics-based stratification model, we showed the corresponding fractions of the predictive variants in each stratified group (Fig. 6d). In general, the fractions of these predictive variants in each risk group exhibits an increased trend from CP to the bone- and liver-MFP groups, featured by the aberration of cell-cycle, p53, and PI3K pathways, indicating a sequential process of malignancy that is concordant with the survival pattern (Fig. 6c). In particular, we noticed that the FGA over 5% is a highly distinguishable feature for CP (31%) versus bone- (88%) and liver-MFP (99%) patients, together with two featuring CNAs: *CDKN1B* deletion and *AR* amplification (Fig. 6d), which is consistent with previous study solely using FGA to predict patient survival of prostate cancer[9]. The predictive and significantly more abundant variant of bone-MFP group than that in liver-MFP group is *SPOP* mutation ($p = 0.0007$, proportion test), which has been found to represent a distinct subtype of prostate cancer that is mutually exclusive to the common E26 transformation-specific (ETS) transcription family fusions[56,57]. The *CDK12* mutation, even though has low fractions in all the three groups (0% in CP, 2.8% in bone-MFP, and 5.9% in liver-MFP), has also been demonstrated to increase genomic instability[58] and aggressiveness[59] in prostate cancer.

The current standard grading system of prostate cancer primarily relies on the Gleason score based on the morphological features of two lesions in histological images. We compared our genomics-based organotropic stratification with the Gleason grading in the TCGA cohort and found a strictly increasing median score of our genomics stratification within each Gleason grade, indicating a high consistency between the two independent systems using genomics and histology (Fig. 6e). In particular, our genomics-based system stratifies more CP patients into the low Gleason-grade group and more liver-MFP patients into the high Gleason-grade group. This suggests that integrating genomic profiles of metastatic prostate cancers to stratify metastatic risks of primary prostate cancer patients can provide an additional dimension for more precise diagnosis and prognosis.

**Discussion**

We developed MetaNet, a computational framework that captures metastatic features within primary tumor genomes to stratify metastatic risk. These features learned from the metastatic tumors can empower MetaNet to sort out the primary cancer patients at high metastatic risk before detection of overt metastasis. Compared to the low-risk group, the high-risk group of patients has turned out to suffer significantly shorter disease-free survival with elevated migratory program significantly enriched in the transcriptome of their tumors. Different from previous studies that identified prognostic genomic biomarkers to predict patient survival[9], MetaNet focused on the biology of metastasis and identified 30 prevalent (fraction > 5%) and significant (FDR < 0.05) variants enriched in organotropic metastasis from a big-data perspective (Fig. 7a). This molecular portrait of organotropic metastasis exhibits strong potential to inform treatment selection (Fig. 5d–f), and surveillance of drug resistance (Fig. 5a–c) and distant metastasis (Fig. 5g–i). In addition, unlike traditional multi-class models that consider the classes to be independent of each other, we proposed the ordinal regression with self-adaptive thresholding to model metastatic dissemination from tissues of origin to proximal sites and distant organs. We demonstrated in prostate cancer that the ordinal regression model with the organotropism-associated variants can predict potential metastatic sites of primary tumors, which stratified the patients into different risk groups with significant differences in survival and histological grades.

Previous studies have proven that tumor molecular features are highly predictive of disease progression and drug response of cancer patients. One longstanding strategy is to develop machine-learning models that learn the likelihood of tumor recurrence or metastasis from a small set of signature genes[4]. This strategy has been commercialized into widely used diagnostic products in breast cancer, such as OncotypeDX[60] and MammaPrint[61]. Unlike this strategy, the contribution of MetaNet lies in the use of somatic variants from a large-scale pan-cancer cohort including 32,176 primary and metastatic samples, the development of machine-learning models using a non-linear classifier with highly informative interpretability, and the clinical application in risk stratification of organ-specific metastases. While direct comparison has not been performed between the two methodologies in a large-scale dataset with genomic and transcriptomic information available, we believe that MetaNet, a genomics-based method, can provide a different perspective complementary to the expression-based assays. And it is anticipatable that a better method might emerge through integrating genomic and transcriptomic data, or even data from digital pathology using complex classifiers like deep learning.

One limitation of our study is that we focused mainly on a small panel of genomic variants (241 genes, Supplementary Data 4). We successfully identified the MFP subtype with worse survival in breast and prostate cancer other than lung and colon cancers. We reasoned that unlike breast cancer and prostate

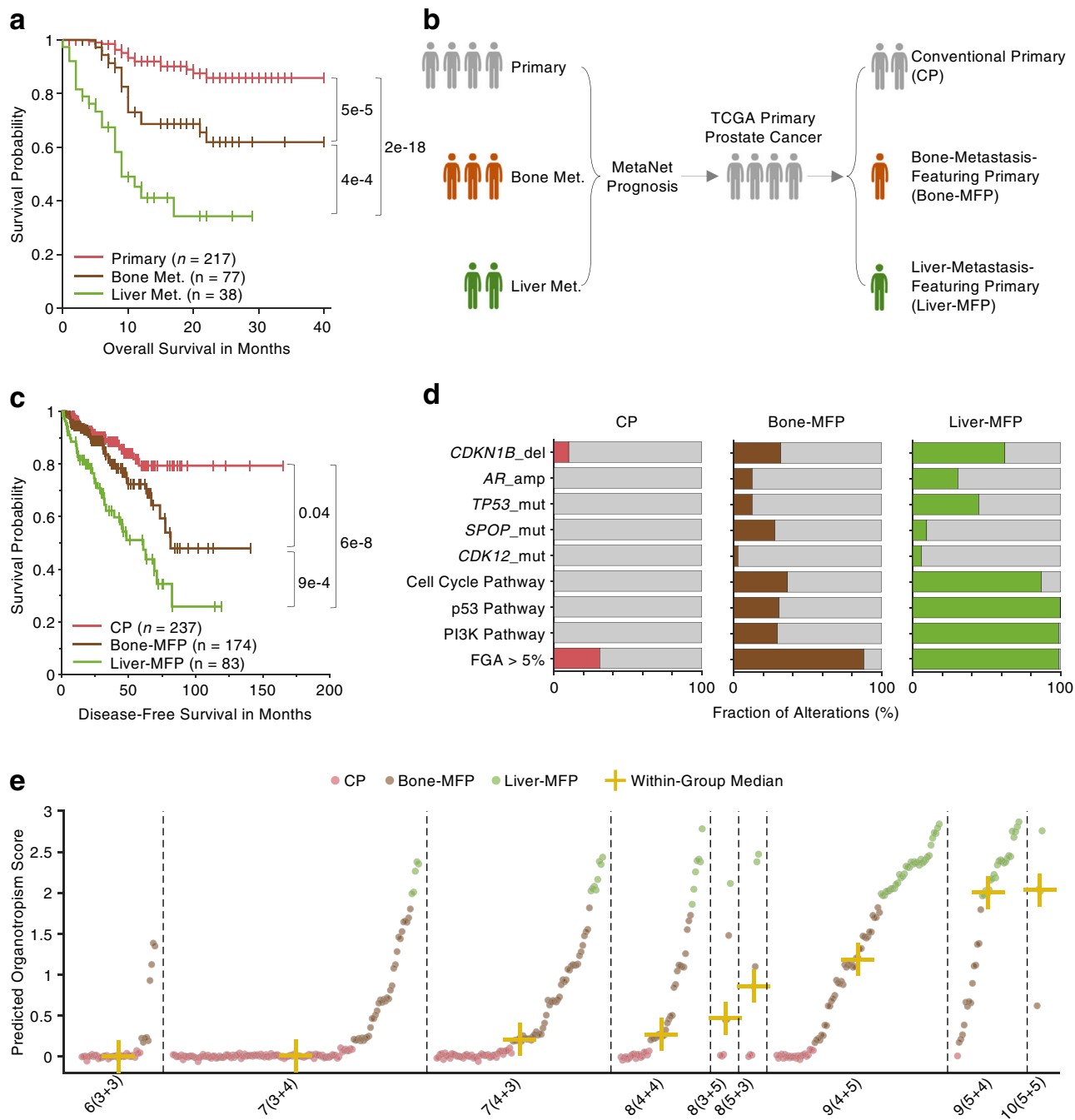

**Fig. 6 Organotropic stratification. a** KM plot represents overall survival differences between primary, bone, and liver metastases of prorate cancer patients in the MSK cohort. The censored data are denoted as + sign. The significance is derived from the two-sided log-rank test. **b** Schematic illustration of MetaNet prognosis module (Model 2) that trains MSK and FMI cohort and validates prognostic power in TCGA cohort by stratifying primary prostate cancer patients into CP, bone-MFP, and liver-MFP groups. **c** KM plot represents DFS differences between CP, bone-MFP and liver-MFP groups of TCGA prorate cancer patients. The censored data are denoted as + sign. The significance is derived from the two-sided log-rank test. **d** Fractions of predictive variants used in Model 2 in CP, bone-MFP, and liver-MFP groups, respectively. Gray color denotes the fraction of wild-type samples in each group. **e** Distribution of organotropic scores (y axis) and stratified groups (node color) of TCGA prostate primary cancer patients in nine different Gleason-grade categories. Each dot denotes the predicted organotropic score of one patient. And the patients are ranked at an ascending order of the predicted organotropic score within each Gleason-grade category. A tiny random number is added into the score for clear visualization of patients with nearly identical scores.

cancer, lung and colon cancers exhibit less genomic difference before and after metastasis (Fig. 2b), leading to poor classification performance (Fig. 2c). To gain a better understanding of the molecular mechanism in these cancers, more efforts should be devoted to investigate evolutionary dynamics of epigenomic

factors and/or tumor microenvironment in the process of cancer cell migration. Many previous studies have shown that transcriptomic and proteomic analyses could reveal the biomarkers directly mediating organotropic metastasis. For example, overexpression of *IL11* and *CTGF* were found to mediate breast

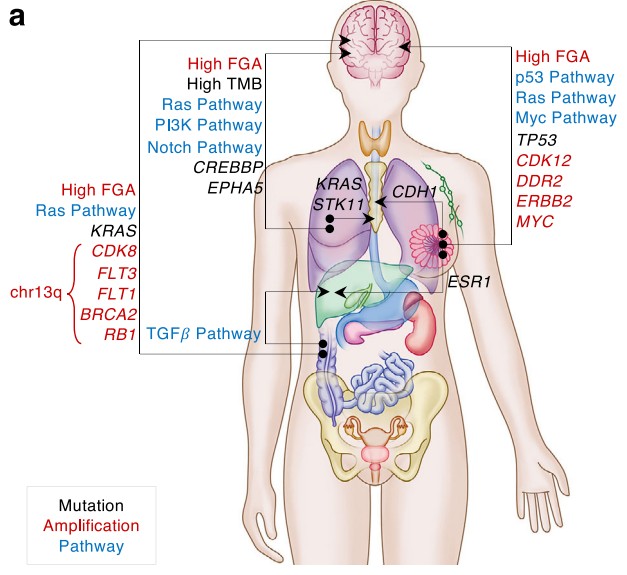

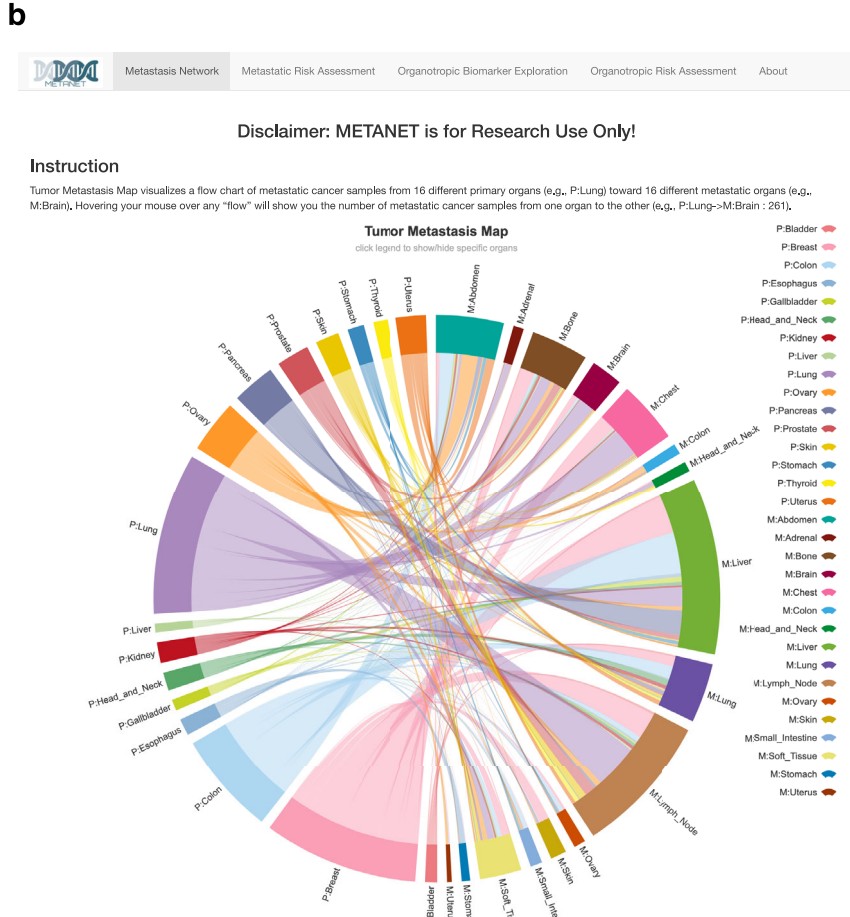

**Fig. 7 Summary of organotropic variants and MetaNet web application. a** 30 abundant (mutant fraction >5% in metastases) and significant (FDR < 0.05, two-sided Chi-squared test, BH correction) organotropic variants/features are shown on human anatomic map. **b** Main page of MetaNet web application displays four functional modules: interactive illustration of metastatic spreading pattern, metastatic risk assessment, organotropic variant exploration, and organotropic stratification. The primary-metastasis identification of unknown primary cancer and tissue of origin prediction of metastatic cancer are under construction.

cancer metastasis to bone[62]. In addition, different exosomal integrins revealed by exosome proteomics were found to associate with organotropic colonization[63]. However, the lack of large-scale transcriptomic and proteomic data in both primary and metastatic samples hinders us to identify the connection from genomics to downstream functional layers that directly dictate organotropic metastasis. We expect that newly emerging high-throughput data and technology, together with sufficiently large data sets gradually accumulated over the years, will soon help close the gap among different types of molecular data and shed light on the entire picture of metastasis biology at a pan-cancer scale.

Another limitation of our study is that we cannot rule out drug resistors from the list of metastasis-related variants based on the present statistical comparison. A better design to overcome this limitation could be the enrollment of patients who did not receive neoadjuvant nor adjuvant chemotherapies. In this case, the statistical comparison of genomic profiles between primary and metastatic samples could unveil metastasis drivers independent of treatment solutions. Further validation through biological experiments is also a must to consolidate the biological roles of the discovered variants.

To enable the wide application of MetaNet by clinic and research communities, we created an R-shiny web application (Fig. 7b) for organotropic biomarker exploration and metastatic risk assessment at https://wanglab.shinyapps.io/metanet. MetaNet has the potential in helping oncologists to assess metastatic risk and relapse time of primary cancer patients, especially to determine whether to surgically resect the tumor given the risk stratification when a biopsy sample is available for sequencing. Finally, even though the present study cannot perform deep investigation and experimental validation for each organotropic biomarker, our online application provides a public window for inquiry of biomarker candidates from other cancer biologists interested in metastatic organotropism.

## Methods

**Data collection of primary and metastatic cancer studies.** The clinical records and tumor genomic sequencing data of primary and metastatic patients from MSK-IMPACT[7], FoundationONE[8], and MET500[64] were collected for this study. In particular, the MSK data were downloaded from cBioPortal (https://www.cbioportal.org/). The FoundationONE data was downloaded from the Genomic Data Commons Data Portal (https://portal.gdc.cancer.gov/). And the MET500 data was downloaded from the official website (https://met500.path.med.umich.edu/). For independent validation, we collected the clinical and genomic sequencing data of four common cancer types: breast, colon, lung, and prostate cancers from TCGA via FireHose data portal (https://gdac.broadinstitute.org/).

**Clinical data profiling of pan-cancer metastasis.** Each primary cancer sample is annotated by its primary site, and each metastatic sample is annotated by its location (metastatic site) and tissue of origin (primary site). The raw clinical record shows that the primary and metastatic sites of the total 32,176 samples are from 360 unique tissues (Supplementary Fig. 1a). For the convenience of downstream study, we merged the tissues into 47 anatomical organs (Supplementary Data 1 and 2) via a computational cancer classification system, OncoTree[65], followed by the consensus of a pathologist panel (Supplementary Fig. 1b). Ruling out the tumors in minor organs (Supplementary Fig. 1b), we visualized the metastatic spreading pattern (Fig. 1a) of all the metastatic tumors from 16 primary sites to 8 metastatic sites using the R packages: circlize[66] and echarter, the R interface of ECharts[67]. We calculated the PCCs of metastatic-site distribution between each pair of primary cancers to construct the PCN ($p < 0.05$, Fig. 1b), and that of primary-site distribution between each pair of metastatic sites to construct the MSN ($p < 0.1$, Fig. 1c), respectively. The network visualization was performed using Cytoscape[68].

**Feature compilation and engineering of genomic variants.** For genomic data compilation and feature engineering, we first merged the gene panels of MSK-IMPACT and FoundationONE to generate an intersect panel of 241 genes (Supplementary Data 4 and Supplementary Fig. 1a). The mutations and copy-number alterations of these 241 genes are used to build a genomic profile of each sample. For the genomic data from MET500 and TCGA generated by whole-exome sequencing, we extracted the mutations and the copy-number alterations of the 241

genes from the whole-exome data in order to unify the genomic profile of the 32,176 samples in total. Each gene was annotated with its chromosome location and total exon size using the GTF file derived from GENCODE[69]. We then engineered extra genomic features: tumor mutation burden (TMB), fraction of genome altered (FGA), chromosome-arm level alteration, and oncogenic pathway aberration. The TMB of each sample, measured at the number of somatic mutations per megabase, was calculated by the total somatic mutation count divided by the total exon size of the 241 genes, followed by a multiplication of 1,000,000. The FGA was calculated by the total size of genes with copy-number alteration normalized by the sum of the chromosomal size of all the 241 genes. We defined and calculated the chromosome-arm level alteration of each sample as over 50% of the genes on the arm have copy-number alteration. Collecting the ten curated oncogenic pathways from TCGA pan-cancer analysis[20], we defined and calculated the aberration of one pathway of each sample if any gene in the pathway has mutation or copy-number alteration.

**Comparison of genomic difference between primary and metastasis.** We performed a proportion test for each variant in the primary and the metastasis of each cancer type, and defined metastasis-enriched variants if the log2 fold change of the variant fraction in the metastasis over that in the primary is larger than 1 and the adjusted $p$-value of the proportion test is <0.05 (Benjamini–Hochberg correction). Conversely, primary-enriched variants were defined in the opposite way.

**Machine-learning model for identification of metastasis-featuring primary tumors.** The training and testing of the machine-learning models were individually performed for each type of the four common cancers: breast, colon, lung, and prostate, using xgboost package in R[27]. The MSK-IMPACT and FoundationONE data were used in the training and testing procedure, while the data from TCGA were used for independent validation. The same features of each sample from different cohorts were compiled and screened for each individual cancer type, especially for those cancer type-specific clinical and histological features (Supplementary Table). Stratified sampling was used to split the samples from MSK-IMPACT and FoundationONE into five folds with identical ratio of primary over metastasis in each fold. One fold was held out for testing, while the other four folds were used to seek the best parameters in a four-fold cross validation. This process was repeated five rounds for each fold of the data so that each sample was tested once to acquire an independent evaluation of metastatic risk, namely Metascore. The model performance was then evaluated by the area under the Receiver Operating Characteristic (ROC) curve using the Metascore of each sample computed in its testing round (Fig. 2c and Supplementary Fig. b1–4). The contribution of each genomic variant to the metastatic risk (Metascore) in each sample was quantified using SHAP value (SHapley Additive exPlanations value[29], Figs. 2d and 3a).

For independent validation, an additional model was trained using all the five-fold data, and then was used to compute the Metascore of TCGA samples (Fig. 3). In order to determine a robust threshold of Metascore to separate the conventional primary group and the metastasis-featuring primary group, three different statistical methods were used:

(a) Lowest-P: Use the Metascore of each patient as the threshold and calculate the $p$-value of log-rank test by comparing the disease-free survival between the two groups, and select the Metascore that yields the lowest $p$-value as the threshold;

(b) Top N%: Rank all the patients based on the Metascore at descending order and select the top 10% of patients as the high-risk group; and

(c) Unsupervised: A Gaussian Mixture Model (GMM) with two components was used to fit the distribution of Metascore, and the threshold was at the intersection of the two components.

The final threshold was determined as the median of the candidate thresholds calculated by the three methods.

**Gene set enrichment analysis.** The Gene Set Enrichment Analysis (GSEA[70]) was performed using GSEAPY, a Python wrapper for GSEA and Enrichr[71]. The 50 hallmark gene sets (h.all.v7.0.symbols.gmt) generated by the Molecular Signature Database[72] were used in the analysis. The permutation was performed within the gene set at 1000 times. The gene list was ranked by the signal-to-noise metric via comparison of the expression in MFP versus CP.

**Survival analysis.** The Kaplan–Meier plot, log-rank test, and the estimation of mean, median, and quantiles of survival time were all performed by MATLAB function MatSurv[73]. The multivariate Cox regression was performed using the R package survival.

**Comparison of genomic difference among different metastatic sites.** We performed a Chi-squared test for the variants of each cancer type in the four common metastatic sites: bone, brain, liver and lung. The regional relapse, i.e., the liver metastasis of liver cancer and the lung metastasis of lung cancer were excluded. The projection of scaled variant fractions into the tetrahedron space was implemented using MATLAB function quatplot3[74].

**ESR1 mutation analysis**. We merged the *ESR1* mutations from MSK, FMI, and MET500 together. The genomic positions in different genomic references were converted into hg19 using LiftOver in the UCSC Genome Browser[75].

**Gene-drug data analysis**. The area under the dose-response curves (AUC) were derived from the original study[50]. To identify the efficacious drugs inhibiting the 23 lung-cancer-brain-metastasis (LUBM) PDCs, we performed a drug-wise standardization of the raw AUCs. To compare the AUCs treated in the 23 LUBM versus the other PDCs, we performed a cell-wise standardization after the drug-wise standardization. Then two-sample *t*-test was performed for the comparison followed by Benjamini–Hochberg correction.

**Machine-learning model for organotropic stratification**. For organotropic stratification, we used the prostate cancer samples from the MSK and FMI cohorts to train an ordinal regression model[76] based on a Proportional Odd Model (POM[77]) using ORCA toolbox[78], a MATLAB framework, and implementation of a wide range ordinal regression methods. Instead of treating each response label (primary, bone metastasis, and liver metastasis) independently using one-hot encoding, we set the label at an order from primary prostate as 0, bone metastasis as 1, to liver metastasis as 2. The genomic variants without enrichment in the primary or organotropic metastases were removed based on our previous enrichment analysis using the $z$-statistic of the proportion test and the Chi-squared statistic. The training was performed using half of the samples and the other half was used in the testing of the accuracy. Independent validation was performed using the TCGA prostate cancer cohort.

**Reporting summary**. Further information on research design is available in the Nature Research Reporting Summary linked to this article.

## Data availability

The published data sets used in this study are listed as follows. The MSK clinical and genomic data of the 10,946 samples were directly downloaded from cBioPortal (https://cbioportal-datahub.s3.amazonaws.com/msk_impact_2017.tar.gz). The clinical and genomic data of the 18,004 samples generated by Foundation Medicine Inc. were accessed from the Genomic Data Commons Data Portal (https://portal.gdc.cancer.gov/) with accession code phs001179. The MET500 clinical and genomic data were directly downloaded from the official website (https://met500.path.med.umich.edu/). And TCGA clinical and genomic data were directly downloaded via FireHose data portal (https://gdac.broadinstitute.org/). All the downloaded genomic data were previously processed by the corresponding data owners, including mutation call table, copy-number alteration table, and gene fusion/rearrangement table. No raw sequencing data were acquired and processed in this study. Anatomic classification of primary and metastatic organs, and supportive data of genomics variants have been provided in Supplementary Data. The remaining intermediate data processed from the above data sets are available from the corresponding author upon request.

## Code availability

The main codes used to develop the MetaNet are available at our GitHub repository: https://github.com/WangLabHKUST/METANET-analysis.

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

## Acknowledgements

We would like to thank Dr. Zheng Hu for sharing the copy-number alteration result from his study. This study was performed using data from MSK-IMPACT released via cBioPortal (ID: msk_impact_2017), MET500 released by University of Michigan (https://met500.path.med.umich.edu/), and FoundationOne released via the GDC portal, Accession phs001179.v1.p1. We would like to extend our sincere gratitude and appreciation to all the data contributors. This work is supported by the Excellent Young Scientists Fund (Hong Kong and Macau) (No. 31922088), grants from the Research Grant Council (C7065-18GF, C4039-19GF, C6021-19EF), Innovation and Technology Commission (ITCPD/17-9, ITS/480/18FP), Department of Science and Technology of Guangdong Province (No. 2020A0505090007), and Hong Kong Epigenomics Project (LKCCFL18SC01-E).

## Author contributions

J.W. conceptualized the project. B.J. carried out the computational studies. B.J. and Q.M. developed the web application. F.Q., X.L., W.X., J.Y., W.F., and Y.C. provided consultancy in medical anatomy. B.J. and J.W. interpreted the data and wrote the manuscript. All authors have read and approved the manuscript.

## Competing interests

The authors declare no competing interests.
