## [Peer Review File · Nature Communications]

REVIEWER COMMENTS

Reviewer #1 (Remarks to the Author): Expert in metastasis and epidemiology

The authors develop a computational framework (termed MetaNet) that integrates clinical and sequencing data from 30K cancers to assess metastatic risk of primary tumors. They state that their program had high accuracy in distinguishing mets from primary breast and prostate cancers. They identify "metastasis featuring primary" (MFP) tumors, a subset of tumors with genomic features "enriched in metastasis." They also identified gen alterations associated with organ-specific mets. The authors state this stratification achieved better prognostic value than standard histological grading, particularly in bone and liver MFP subtypes.

This is a very important study idea that seeks to improve the staging/prognostication of metastatic cancer. There are a ton of data, and I think one of the most important parts in the interpretation of the work is the biostatistics and analysis. I hope to provide comments that will improve the quality of the work.

I have several major comments:

(1) In the grouping of cancers (Supplementary Table 1), I am a little confused why certain cancers/locations are grouped together. Grouping cancers in this type of work is very important because the drainage/metastatic patterns of certain cancers are very unique and should not be confused with other cancers. Can the authors please clarify:
Why are endometrial and cervical cancer grouped together? These are very different entities that affect very different types of patient populations.
Why are so many abdominal malignancies grouped together? I would exclude many of these from the analysis, unless the authors know what they are. For example, an abdominal "mass" could be a primary sarcoma, a colorectal cancer, a pancreatic cancer, a metastasis from any type of abdominal/pelvic malignancy (including more common cancers like prostate, bladder, rectum). Ascites is likely to be metastatic deposit from something like ovarian cancer. Groin is for some reason part of abdomen, but I would think a "groin mass" differential diagnosis would be metastasis from a penile/vulvar cancer, lymphoma, or a few other malignancies, but not related to pancreas/colorectum. Why is "axilla" part of chest? The number one cancer in the axilla is probably a nodal metastasis from breast cancer, followed by a met from another location, followed by lymphoma. Why is "chest wall" part of chest? I would think this is probably a breast cancer/recurrence or perhaps a skin cancer like melanoma?
Similarly, "mediastinal mass" and "mediastinum" are part of "Chest". These should likely be considered either with a primary lung cancer or a lymphoma (These are the most common diagnoses). It doesn't make sense to me to group these with "Chest wall". Pleura is also grouped with chest, but I would think this should be grouped with lung, because the most common tumor here is a mesothelioma. "Chest" typically implies something in the chest wall or breast cancer.
How do the authors know that "pelvis" is part of the bone and not within the pelvis itself? If within the pelvis, then this is likely a cancer like prostate, endometrium, cervix, etc.
Penis should not be part of the "pelvic cavity" -- it's technically outside of the cavity, and it is more likely to metastasize to the groin lymph nodes. In some ways, it is related to urethra.
"Periarotic mass" sounds like a lymph node metastasis, and this could be from malignancies like prostate, endometrial, cervix, kidney, bladder. I would NOT think that this is related to a "Soft tissue" tumor like the authors have.
"Ureter" and "upper tract" are probably urothelial cancers, which is similar to bladder cancer. I would not group these with urethral cancers, which have a different metastatic pattern, typically to the groin. Ureter and upper tract typically metastasize to abdominal LNs and the lungs.
Given these discrepancies, I wonder if the authors should have primary clinicians on the author line?

(2) The term MFP is novel, and it is great that the authors are finding new types of cancers that are

likely to harbor metastases. They highlight that their new method is more prognostic for prostate and breast cancers; however, in reviewing Sup Figure 2, it looks like the system did not add any diagnostic value for breast cancer or colon cancer, and it had minimal improvement for lung cancer. Breast, colon, and lung cancers make up ~60-70% of cancer diagnoses in developed countries, so the conclusion I would draw by looking at these data is that the system does not add much diagnostic value. Can the authors please clarify this? Was it helpful for other cancers as well? Also, for cancers like prostate, how does this novel system compare to something like the genomic classifier (GC)? And what about other mutations that are mentioned in NCCN guidelines, like BRCA, ATM, PALB2, CHEK2, MLH, MSH2.

(3) For many clinicians, myself included, machine learning is a black box, and it is one that clinicians can never use. There is the stereotypical figure (Figure 6b) of data going in, "learned features" and then some sort of risk calculated (Figure 6). How does the machine actually do this? Is machine learning superior to other methods, eg logistic regression, random forest? And could readers use this work? It would be great if a clinician like myself could. Currently, as a reader of this work, it appears the authors have a good model, but it is unclear how it works, and I don't know how I can apply it to my own practice (though I would really like to).

(4) Why are there no coauthors from the affiliated US institutions? Did the institutions approve of this work?

Minor comments:

I would integrate epidemiology of metastases into the discussion. For example, these have recently been published: PMID 32562887, 32363344

Other comments:

- What are the noteworthy results?

Novel staging system for metastatic cancer, though some details need more discussion, as above

- Will the work be of significance to the field and related fields? How does it compare to the established literature? If the work is not original, please provide relevant references.

Yes, this could be novel and significant, but I don't think it is in the current form.

- Does the work support the conclusions and claims, or is additional evidence needed?

Unclear, as above.

- Are there any flaws in the data analysis, interpretation and conclusions? - Do these prohibit publication or require revision?

Please see above.

- Is the methodology sound? Does the work meet the expected standards in your field?

I recommend having other clinicians review the grouping system, and having machine learning experts review the methods.

- Is there enough detail provided in the methods for the work to be reproduced?

As above

Reviewer #2 (Remarks to the Author): Expert in machine learning and genomics

Metastasis is the leading cause of cancer death. Based on big data integration, Jiang et al. developed a machine learning method named MetaNet to discover the underlying patterns of pan-cancer metastasis and to predict patient prognosis based on newly defined "metascore" and "organotropic propensities". The authors have successfully integrated two major datasets with genomic data from over 30,000 cancer patients and found that multimodal data integration can dramatically improve the prediction of cancer metastasis as well as patient survival. Overall, this study is highly innovative, and

it will provide novel insights to the fields of cancer diagnosis and cancer genomics. The manuscript is in general well written, and the figures are well organized. Yet there are some minor suggestions for improving the manuscript.

1. The identification and characterization of Metastasis-Featuring Primary in breast cancer is very impressive but the same strategy on other cancer types (such as prostate cancer) has not been carefully discussed and compared. It will be helpful to include a paragraph to discuss different factor that might cause the differences among various cancer types;
2. SHAP value is an interesting metric to highlight features but it is not obvious to most of the audience, especially those from biology or medicine backgrounds. The meaning of this value should be further described when it was first introduced;
3. In the MetaNET web server, the "About" page should provide more details and be consistent with the manuscript;
4. In the MetaNET web server, the "Biomarker Exploration" page should provide a suggested list of significant biomarkers for each cancer type, in order to make it more user-friendly;
5. In the MetaNET web server, the input options should allow "missing value", as clinical information is frequently missing in real-world practice;

In summary, the manuscript is suitable for publication after minor revision.

Reviewer #3 (Remarks to the Author): Expert in metastasis and cancer genomics

This manuscript describes the genomic analysis of 241 genes across 32,176 primary tumor and metastasis samples from four different studies. The authors compare the mutation and copy number variation data of the metastases with unmatched primary tumors to attempt to find mutational signatures that would enable metastasis prognosis using primary tumor analysis. Moreover, the authors explore organotropism of the various primary tumors in this data, as well as develop algorithms that predict metastasis incidence and target organ.

The question that the authors address is an important one, since as they note, the majority of mortality in solid tumors is associated with metastatic disease rather than the primary tumor. However, a number of concerns exist about this manuscript. First, a minor concern is that this manuscript would benefit from editing for English. Second, and most importantly, much of the analysis presented in this manuscript is already known, and some of it very well established in metastasis research. For example, the fact that different tumor types reproducibly seed specific target organs was first published by Paget in the late 1800s. The author's suggestion that seeding might be partially explained by blood flow patterns was initially suggested by Ewing, approximately 100 years ago. ESR1 mutations have been associated with metastasis in breast cancer in a number of manuscripts in recent years. The association of chromosomal instability with metastatic spread has also been previously published. The role of KRAS mutations in colorectal cancer metastasis has been the subject of a number of recent publications. So while the authors have done a nice job presenting the data in graphic form, it is difficult to identify the novel aspects that may be in this manuscript.

Reviewer #4 (Remarks to the Author): Expert in machine learning and cancer genomics

Summary of the work

The authors have developed a computational framework (MetaNet) that integrates clinical and sequencing data from 32,176 primary and metastatic cancer cases, to assess metastatic risks of primary tumors. Using this framework, the authors identified Metastasis-Featuring Primary (MFP) tumors, a subset of primary tumors with genomic features enriched in metastasis, and demonstrated their high metastatic risks with significantly shorter disease-free survivals and higher migratory potential. The MSK-IMPACT and FoundationONE data were used in the training and testing procedure,

while the data from TCGA were used for independent validation. It should be noted that the machine learning aspect of the work is not novel but the application to an outstanding problem could lead to novel findings. Furthermore, the authors should be commended on the extent of the analysis that they have performed.

Major comments

1- The Introduction suffers from a poor overview and does not properly represent the contents of the paper. Introduction is sorely lacking references to others who have performed similar work before, and needs to address the significance of the novelty of this investigation (it could be quite significant, but is not conveyed as such). The introduction needs to spend much more time explaining what exactly this investigation is doing, and address significant background which is overlooked or assumed by the authors. The introduction is hampered by strong grammatical mistakes which cause excessive confusion to the reader.

2- The authors hypothesize that genomic variations of primary tumors could be used as indicators for metastatic risk assessment. However, currently available genomic databases have only a small number of paired primary/metastatic samples, which are insufficient to sort out reliable prognostic biomarkers applicable in a larger cancer population. However, the authors have proposed to use unpaired sample data (derived from different patients) to characterize tumor aggressiveness. Using a computational pipeline, the authors train a classifier to differentiate between primary and metastasis and then try to link the findings to clinical outcome.

In my opinion, the authors should definitely try to directly link the cases to outcome (as the final end-point) rather than using metastasis as an intermediate end-point. There are well-characterized algorithms that directly link features (genomics variations in this case) to outcome data (e.g., overall survival, disease free survival). Examples are cox-regression, cox-lasso, survival random forests, etc. Without this analysis, the reviewer believes that we cannot fairly assess the contribution of the proposed computational approach (i.e., using metastasis as the end-point and then link to survival versus directly linking genomic variation to survival).

3- On Page 28, the authors state that "A threshold was selected to achieve the best separation of conventional primary group and metastasis-featuring primary group using disease-free survival time in TCGA." This reviewer believes that the thresholds cannot be set based on the "unseen test data". The authors need to select the thresholds based on the training and/or validation data and should NOT touch the test set (TCGA data) to achieve the best results. This is a major concern that makes it difficult to assess the validity of the findings.

4- The paper lack details to reproduce the results. Given that "Machine learning" is incorporated in the title of the paper, I highly suggest that the authors make sure that they include enough details to enable others to reproduce results – this could be in the form of releasing code as well.

Conclusion

Taken together, while commending the extent of the work that has been performed, this reviewer believes that the manuscript requires substantial improvement to allow a fair assessment of the findings and contributions.

Point-by-point Response* to Reviewers' Comments#

*Sentences excerpted from the manuscript are marked in blue

#All comments from reviewers are cited in quoted format

Response to Reviewer #1: Expert in metastasis and epidemiology

General comments

“The authors develop a computational framework (termed MetaNet) that integrates clinical and sequencing data from 30K cancers to assess metastatic risk of primary tumors. They state that their program had high accuracy in distinguishing mets from primary breast and prostate cancers. They identify “metastasis featuring primary” (MFP) tumors, a subset of tumors with genomic features “enriched in metastasis.” They also identified gen alterations associated with organ-specific mets. The authors state this stratification achieved better prognostic value than standard histological grading, particularly in bone and liver MFP subtypes. This is a very important study idea that seeks to improve the staging/prognostication of metastatic cancer. There are a ton of data, and I think one of the most important parts in the interpretation of the work is the biostatistics and analysis. I hope to provide comments that will improve the quality of the work.”

Author response:

We thank the reviewer for the summary of our work.

Comment 1.1:

“In the grouping of cancers (Supplementary Table 1), I am a little confused why certain cancers/locations are grouped together. Grouping cancers in this type of work is very important because the drainage/metastatic patterns of certain cancers are very unique and should not be confused with other cancers. Can the authors please clarify: Why are endometrial and cervical cancer grouped together? These are very different entities that affect very different types of patient populations.”

Author response:

We appreciate this professional comment from the reviewer. In the original manuscript, we grouped the cancer locations at the level of general anatomic organs. In this revision, we follow the reviewer’s comment and re-group tumor tissues by considering additional information such as histology and tissue of origins. The re-grouping sorts out “*cervical cancer*” as a separate group from the original “Uterus” group (see the table below).

Original Annotation	Grouped Term	No. Samples
Cervix Uteri	Cervix	134
Cervix	Cervix	54
Mullerian	Uterus	1
Uterus, NOS	Uterus	368
Uterus	Uterus	324

All relevant analyses have been updated according to this revision. Note that the re-grouping in this revision does **not** change the major conclusions of our original manuscript.

Reflected in the revised manuscript:

We have updated the re-grouping of 29,450 cancer samples in the updated **Supplementary Tables 1 and 2** for the primary tumor sites and the metastatic sites, respectively. And accordingly, we have regenerated Figures 1a-c, 4a-d, 5g-h, and Supplementary Figures 1a, 4a-e, 5a, 6a and 6g based on the updated re-grouping results.

Comment 1.2:

“Why are so many abdominal malignancies grouped together? I would exclude many of these from the analysis, unless the authors know what they are. For example, an abdominal “mass” could be a primary sarcoma, a colorectal cancer, a pancreatic cancer, a metastasis from any type of abdominal/pelvic malignancy (including more common cancers like prostate, bladder, rectum). Ascites is likely to be metastatic deposit from something like ovarian cancer. Groin is for some reason part of abdomen, but I would think a “groin mass” differential diagnosis would be metastasis from a penile/vulvar cancer, lymphoma, or a few other malignancies, but not related to pancreas/colorectum.”

Author response:

We agree that the abdominal malignancies might originate from different tissue of origins. And hence, we accept the suggestion by the reviewer and “exclude” those malignancies from our downstream analysis.

Reflected in the revised manuscript:

We put the metastatic malignancies with arbitrary tissues of origins located in abdominal, chest and pelvic cavities, and the ones without explicit locations at soft tissues, skin and head and neck, and the ones with small sample sizes located at stomach and small intestine, into a miscellaneous group (Misc.), as shown in **Figure 1a**. These samples are also “excluded” from the analysis on the prevalence of organotropism in **Figure 4a**.

Comment 1.3:

“Why is “axilla” part of chest? The number one cancer in the axilla is probably a nodal metastasis from breast cancer, followed by a met from another location, followed by lymphoma. Why is “chest wall” part of chest? I would think this is probably a breast cancer/recurrence or perhaps a skin cancer like melanoma? Similarly, “mediastinal mass” and “mediastinum” are part of “Chest”. These should likely be considered either with a primary lung cancer or a lymphoma (These are the most common diagnoses). It doesn’t make sense to me to group these with “Chest wall”. Pleura is also grouped with chest, but I would think this should be grouped with lung, because the most common tumor here is a mesothelioma. “Chest” typically implies something in the chest wall or breast cancer.”

Author response:

We thank the reviewer to point out the issue of the original version. Accordingly, we “exclude” the malignancies located in chest cavity without clear annotation of tissue of origin from downstream analysis.

Comment 1.4:

How do the authors know that “pelvis” is part of the bone and not within the pelvis itself? If within the pelvis, then this is likely a cancer like prostate, endometrium, cervix, etc. Penis should not be part of the “pelvic cavity” -- it’s technically outside of the cavity, and it is more likely to metastasize to the groin lymph nodes. In some ways, it is related to urethra. “Periarotic

mass” sounds like a lymph node metastasis, and this could be from malignancies like prostate, endometrial, cervix, kidney, bladder. I would NOT think that this is related to a “Soft tissue” tumor like the authors have. “Ureter” and “upper tract” are probably urothelial cancers, which is similar to bladder cancer. I would not group these with urethral cancers, which have a different metastatic pattern, typically to the groin. Ureter and upper tract typically metastasize to abdominal LNs and the lungs.

Author response:

Suggested by the reviewer, we also “*exclude*” the malignancies located in pelvic cavity without clear annotation of tissue of origin from downstream analysis.

Comment 1.5:

“Given these discrepancies, I wonder if the authors should have primary clinicians on the author line?”

Author response:

In the original manuscript, the cancer grouping was mainly performed by two postgraduate students, Weiqi Xu and Fufang Qiu. In the revision, considering the complexity of pan-cancer medicine, we further invited primary clinicians, including Dr. Yong Cao and Dr. Weilun Fu with expertise in brain disorders from Beijing Tiantan Hospital, Dr. Xuefeng Li with expertise in thoracic disorders from Guangzhou Medical University, and Dr. Jun Yu with expertise in digestive diseases from Prince of Wales Hospital, to conduct the regrouping of cancer samples.

Comment 2:

“The term MFP is novel, and it is great that the authors are finding new types of cancers that are likely to harbor metastases. They highlight that their new method is more prognostic for prostate and breast cancers; however, in reviewing Sup Figure 2, it looks like the system did not add any diagnostic value for breast cancer or colon cancer, and it had minimal improvement for lung cancer. Breast, colon, and lung cancers make up ~60-70% of cancer diagnoses in developed countries, so the conclusion I would draw by looking at these data is that the system does not add much diagnostic value. Can the authors please clarify this? Was it helpful for other cancers as well? Also, for cancers like prostate, how does this novel system

compare to something like the genomic classifier (GC)? And what about other mutations that are mentioned in NCCN guidelines, like BRCA, ATM, PALB2, CHEK2, MLH, MSH2.”

Author response:

We thank the reviewer for raising the questions regarding what types of cancers exist identifiable MFP, and whether the mutations mentioned in NCCN are predictive of metastatic risk as well.

First, we would like to clarify that, whether our system can add extra diagnostic value depends on whether the system can identify the MFP subtype. We showed in **Sup Fig. 2** and equivalently **Fig. 2c** that using the metric of AUROC, our model with genomic features (red color) outperformed the base models with clinical and histological features but without the genomic features (dark blue) by 38% in prostate cancer (from 0.58 to 0.8), 16% in lung cancer (from 0.57 to 0.67), 8% in breast cancer (from 0.76 to 0.82) and 1% in colon cancer (from 0.556 to 0.562). Using the system, we further successfully identified the MFP subtype with shorter DFS than the Conventional Primary (CP) subtype in TCGA cohorts of breast cancer ($p < 0.0001$, log-rank test, **Fig. 3d**), prostate cancer ($p = 0.009$, log-rank test, **Sup Fig. 3d**) and lung cancer ($p = 0.28$ in LUAD and $p = 0.19$ in LUSC, log-rank test, **Sup Fig. 3e**). Although the AUROC was improved by only 8% in breast cancer (diagnostic value, **Sup Fig. 2b1**), our model still successfully identified the MFP subtype (prognostic value, $p < 0.0001$, **Fig. 3d**), and even in different histological subtypes (**Sup Fig. 3f**). Regarding the lung and colon cancers, we agree with the reviewer that our system does not add much diagnostic value. We believe it might reflect the intrinsic properties of these two tumor types. The following statement has been added in the **Discussion**:

“We successfully identified the MFP subtype with worse survival in breast and prostate cancer other than lung and colon cancers. We reasoned that unlike breast cancer and prostate cancer, lung and colon cancers exhibit less genomic difference before and after metastasis (Fig. 2b), leading to poor classification performance (Fig. 2c). To gain better understanding of the molecular mechanism in these cancers, more efforts should be devoted to investigate evolutionary dynamics of epigenomic factors and/or tumor microenvironment in the process of cancer cell migration.”

Second, whether the MFP is identifiable in other cancers depends on the sufficiency of the genomic and clinical data, and whether there exists genomic difference between metastatic and primary tumor genomes. From the following table, we listed the data size of primary and metastatic samples in each cancer type, and showed that only lung, breast and colon cancers have sufficient sample sizes over 1,000 in both primary and metastatic samples. We think a sample size over 1,000 could allow us to train a robust model.

Primary Site	No. Primary	No. Metastasis
Lung	2993	1701
Breast	1637	1627
Colon	1575	1174
Pancreas	579	523
Liver	517	89
Prostate	502	438
Bladder	482	134
Kidney	463	215
Stomach	416	124
Esophagus	374	123
Skin	290	247
Uterus	288	199
Ovary	285	360
Head and Neck	219	218
Gallbladder	121	117
Thyroid	103	153

We further considered prostate cancer because we found in **Fig. 2b** that the most significant genomic difference between primary and metastatic tumors is identified in prostate cancer via statistical testing. Finally, we chose breast, prostate, lung and colon cancers to conduct our proof-of-concept study for the MFP concept. For other cancer types, the MFP cases might exist but is challenging to be identified due to the insufficiency of samples and the dimensional diversity of molecular features. We think our system may work in other cancer types as well once we can collect sufficient samples and/or other types of high-throughput molecular features in transcriptome or epigenome.

The following statement has been added in the **Discussion**:

“We expect that newly emerging high throughput data and technology, together with sufficiently large datasets gradually accumulated over the years, will soon help close the gap

among different types of molecular data and shed light on the entire picture of metastasis biology at pan-cancer scale.”

Third, we have tested the model using only the 7 genes (*BRCA1/2*, *ATM*, *PALB2*, *CHEK2*, *MLH*, *MSH2*) mentioned in the NCCN guideline and the classification performance metric AUROC is only 0.58 (see the red dashed line in the figure below). Although the germline mutations in the 7 genes are believed to be predisposing factors of prostate cancer, our result demonstrated that the 7 NCCN genes do not contribute to the classification of primary and metastatic prostate cancers.

We further visualize the fraction of mutant samples of the 7 genes in prostate cancer, as shown in the figure below. The *MLH1* is the only gene whose mutant fraction is significant different between primary and metastatic prostate cancers ($p = 0.02$, proportion test). We reason that this limited genomic difference is why they are not predictive of metastatic risk.

Comment 3:

“For many clinicians, myself included, machine learning is a black box, and it is one that clinicians can never use. There is the stereotypical figure (Figure 6b) of data going in, “learned features” and then some sort of risk calculated (Figure 6). How does the machine actually do this? Is machine learning superior to other methods, eg logistic regression, random forest? And could readers use this work? It would be great if a clinician like myself could. Currently, as a reader of this work, it appears the authors have a good model, but it is unclear how it works, and I don’t know how I can apply it to my own practice (though I would really like to).”

Author response:

As described in the Section of **Organotropic stratification of primary tumors**, and the **Methods** Section, the machine-learning model shown in **Figure 6b** is built upon the framework of ordinary regression that integrates genomic features to estimate metastatic risk from the tissue of origin toward proximal or distant organs. Generally, this model is to learn a genomic score from training data, and project it into a risk score via a link function $f()$, as shown below:

$$\text{risk_score} = f(\text{genomic_score})$$

where $f()$ is a function linking the genomic score and risk score, and the genomic score is defined as the linear combination of genomic features, such as the overall fraction of altered genome, the alteration/activity of various pathways including PI3K pathway, p53 pathway, cell cycle pathway, mutations and/or copy number alterations of genes such as CDK12, SPOP, TP53, AR and CDKN1B. The risk score, a.k.a., organotropism score in **Figure 6e**, approximately ranges from 0 to 3 in the independent evaluation of the 500 patients from TCGA-prostate cohort. And the model automatically estimates the thresholds among three different stratified groups (**Figure 6e**):

- (1) Conventional Primary (CP) Group: organotropism score less than 0.1568,
- (2) Bone-Metastasis-Featuring Primary (Bone-MFP) Group: organotropism score between 0.1568 and 1.8058,
- (3) Liver-Metastasis-Featuring Primary (Liver-MFP) Group: organotropism score larger than 1.8058.

For the question in regarding the superiority of our method to other methods, e.g., logistic regression (LR) and random forest (RF), we have conducted a fair comparison of our method, LR and RF using the same dataset in **Figure 6**. Training and validating the LR and RF models,

we have shown in the following figure that compared to the prognostic performance of our model shown in **Figure 6c** (the same as the left panel), the LR (middle panel) and RF (right panel) fail to stratify the patients from TCGA-prostate cohort into Bone-MFP and Liver-MFP groups with significant difference in DFS (the insignificant p values are highlighted in red).

The reason why our new model outperforms the other two methods is because it adds more penalty on the misclassifications between CP vs. Liver-MFP than those between CP vs. Bone-MFP and Bone-MFP vs. Liver-MFP, ensuring the three labels, CP, Bone-MFP and Liver-MFP, are ordinal rather than three independent ones. A description in the **Discussion** Section about the difference between our model and other typical machine-learning models has been added for non-machine-learning experts to understand the mechanism:

“In addition, unlike traditional multi-class models that consider the classes to be independent of each other, we proposed the ordinal regression with self-adaptive thresholding to model metastatic dissemination from tissues of origin to proximal sites and distant organs. We demonstrated in prostate cancer that the ordinal regression model with the organotropism-associated variants can predict potential metastatic sites of primary tumors, which stratified the patients into different risk groups with significant differences in survival and histological grades.”

In addition, as mentioned in the original manuscript, we developed a web application, namely METANET, at <https://wanglab.shinyapps.io/metanet> to encourage readers to apply the model in their practice. For more details about how to use our web application, please visit the “Tutorial” page on the main page.

Comment 4:

“Why are there no coauthors from the affiliated US institutions? Did the institutions approve of this work?”

Author response:

We described in the Subsection, **Data collection of primary and metastatic cancer studies**, about how we acquired the clinical and genomic data in a legal way following the corresponding regulations. For MSK data, we found that the MSK group generously offered the data access with the license stated as: “The data are available under the ODC Open Database License (ODbL)(<http://opendatacommons.org/licenses/odbl/1.0/>) (summary available here: <http://www.opendatacommons.org/licenses/odbl/1-0/summary/>): you are free to share and modify the data so long as you attribute any public use of the database, or works produced from the database; keep the resulting data-sets open; and offer your shared or adapted version of the data-set under the same ODbL license.” Here we followed this rule to share our datasets and software freely at <https://wanglab.shinyapps.io/metanet>. For MET500 data collected and maintained by the University of Michigan, we followed the Web Site Terms and Conditions of Use listed on their website: <https://met500.path.med.umich.edu/terms>. And for the Foundation Medicine data, we applied the data access to phs001179.v1.p1 from dbGap using project #14905 (a start-up fund of the corresponding author at the Hong Kong University of Science and Technology). According to the data use policy, FoundationOne allows academic publication if the paper acknowledged the data contributors. Overall, in this whole project we have not downloaded the raw sequencing data of these three cohorts, and hence, we would not be able to identify any patients from their genetic markups.

Reflected in the revised manuscript:

We have added the following sentences in the **Acknowledgements**:

“This study was performed using data from MSK-IMPACT released via cBioPortal (ID: msk_impact_2017), MET500 metastatic cancer cohort released by the University of Michigan (<https://met500.path.med.umich.edu/>), and FoundationOne released via the GDC portal, Accession phs001179.v1.p1. We would like to extend our sincere gratitude and appreciation to all the data contributors.”

Minor comments:

I would integrate epidemiology of metastases into the discussion. For example, these have recently been published: PMID 32562887, 32363344

Author response:

We thank the reviewer for providing us these two highly relevant studies.

Reflected in the revised manuscript:

We have integrated these two studies into our **Introduction** where we ever cited some major facts in metastatic organotropism discovered by previous studies.

“Epidemiological studies have discovered that depending on the tissue of origins and other factors, metastatic tumor cells have preference to seeding at certain distant organs, known as organotropism ^{11,12}. For example, the epidemiological data from 2010 to 2015 have shown that approximately 80% of synchronous brain metastases originated from lung primaries ¹³. Metastases from the same tissue but colonizing at different organs may result in different survivals ^{14,15}. In particular, hepatitis metastases, as the most prevalent diagnosis, lead to significantly worse clinical outcome than non-hepatic metastases in most cancer types ¹⁶.”

Reviewer #2 (Remarks to the Author): Expert in machine learning and genomics

General comments

“Metastasis is the leading cause of cancer death. Based on big data integration, Jiang et al. developed a machine learning method named MetaNet to discover the underlying patterns of pan-cancer metastasis and to predict patient prognosis based on newly defined “metascore” and “organotropic propensities”. The authors have successfully integrated two major datasets with genomic data from over 30,000 cancer patients and found that multimodal data integration can dramatically improve the prediction of cancer metastasis as well as patient survival. Overall, this study is highly innovative, and it will provide novel insights to the fields of cancer diagnosis and cancer genomics. The manuscript is in general well written, and the figures are well organized. Yet there are some minor suggestions for improving the manuscript.”

Author response:

We thank the reviewer for the summary of our work.

Minor suggestion 1:

“The identification and characterization of Metastasis-Featuring Primary in breast cancer is very impressive but the same strategy on other cancer types (such as prostate cancer) has not been carefully discussed and compared. It will be helpful to include a paragraph to discuss different factor that might cause the differences among various cancer types.”

Author response:

We thank the reviewer for this useful suggestion, and we did have discussion in the **Results** Section about the identification and characterization of the MFP subtypes in breast, prostate, lung, and colon cancers. We showed that the model could distinguish metastases from primaries well in breast and prostate cancers, other than lung and colon cancers (Figure 2c), which could be explained by the genomic similarity between primary and metastatic cancers. Genomically similar cancers, even though at different locations, are difficult to be distinguished by our model as in the case of colon and lung cancers. We further pointed out that the high genomic similarity between primary and metastatic colon and lung cancers indicated a shorter sampling interval between the two tumors (equivalently as disease-free survival) than those in breast and prostate

cancers. Assuming a constant mutation rate, we concluded that shorter disease-free survival made the metastatic tumor genome similar to the primary one.

Reflected in the revised manuscript:

“This result demonstrated that the primary and metastatic breast and prostate tumors are genomically different, while in lung and colon cancer the genomes are alike, which is similar with our observation in the comparison of the genomic profiles (Fig. 2b). From an evolutionary perspective, it suggests that unlike lung and colon cancers, breast and prostate cancers may follow certain evolutionary modes in which only novel clones resistant to hormone treatments can thrive in the metastasis. In terms of clinical implication, disease-free survivals of breast and prostate cancer patients are generally longer than those with lung and colon cancers²⁶, during which the metastases of breast and prostate cancers have longer time to evolve and acquire more variants than those of lung and colon cancers under the assumption of constant mutation rate.”

Minor suggestion 2:

“SHAP value is an interesting metric to highlight features but it is not obvious to most of the audience, especially those from biology or medicine backgrounds. The meaning of this value should be further described when it was first introduced.”

Author response:

We thank the reviewer for this useful suggestion, and we have added a description about SHAP value in the main text.

Reflected in the revised manuscript:

“To further understand what genomic variants are used in the model for metastatic risk prediction, we used SHapley Additive exPlanations (SHAP) value²⁵ to untangle the tree-based model by visualizing gene-wise contribution to the metastatic risk of breast cancer (Fig. 2d). A positive SHAP value indicates that the genomic feature has a positive contribution to the metastatic risk, while a negative value represents a negative impact on the risk.”

Minor suggestion 3:

“In the MetaNET web server, the “About” page should provide more details and be consistent with the manuscript.”

Author response:

We thank the reviewer for this useful suggestion, and we have revised the “About” webpage for the consistency.

Minor suggestion 4:

“In the MetaNET web server, the “Biomarker Exploration” page should provide a suggested list of significant biomarkers for each cancer type, in order to make it more user-friendly.”

Author response:

We thank the reviewer for this useful suggestion, and we have added a list of significant biomarkers onto the “Biomarker Exploration” page.

Minor suggestion 5:

“In the MetaNET web server, the input options should allow “missing value”, as clinical information is frequency missing in real-world practice.

Author response:

In fact, our Model 1 on the page of “Metastatic Risk Assessment” at our webserver considered three types of “clinical information” as inputs: Age, Race and Histologic Subtype. We provided “Unknown” option for the inputs of Race and Histologic Subtype. For Age, the server will consider the input as “Unknown” if users input nothing in the box. And we provided user a hint in the blank box that the mean age of our investigated cohort is 53 years old. In our Model 2 on the page of “Organotropic Risk Assessment”, the model does not rely on any clinical information.

Conclusion:

In summary, the manuscript is suitable for publication after minor revision.”

Reviewer #3 (Remarks to the Author): Expert in metastasis and cancer genomics

General comments:

“This manuscript describes the genomic analysis of 241 genes across 32,176 primary tumor and metastasis samples from four different studies. The authors compare the mutation and copy number variation data of the metastases with unmatched primary tumors to attempt to find mutational signatures that would enable metastasis prognosis using primary tumor analysis. Moreover, the authors explore organotropism of the various primary tumors in this data, as well as develop algorithms that predict metastasis incidence and target organ. The question that the authors address is an important one, since as they note, the majority of mortality in solid tumors is associated with metastatic disease rather than the primary tumor. However, a number of concerns exist about this manuscript.”

Concern 1:

“First, a minor concern is that this manuscript would benefit from editing for English.”

Author response:

We have carefully revised the manuscript and double checked the grammatic mistakes.

Concern 2:

“Second, and most importantly, much of the analysis presented in this manuscript is already known, and some of it very well established in metastasis research. For example, the fact that different tumor types reproducibly seed specific target organs was first published by Paget in the late 1800s. The author’s suggestion that seeding might be partially explained by blood flow patterns was initially suggested by Ewing, approximately 100 years ago. ESR1 mutations have been associated with metastasis in breast cancer in a number of manuscripts in recent years. The association of chromosomal instability with metastatic spread has also been previously published. The role of KRAS mutations in colorectal cancer metastasis has been the subject of a number of recent publications.”

“So while the authors have done a nice job presenting the data in graphic form, it is difficult to identify the novel aspects that may be in this manuscript.”

Author response:

We are sorry for unclear description in the original version which might cause some confusions. We did highly appreciate the previous studies achieved by many great scientists and cited those works accordingly in our original manuscript, including Paget's finding in 1889, the "seed and soil" hypothesis. However, although we, for the first time, observed consistent findings with those classical works from a big-data perspective (over 30k patients), we did not claim that these observations were the novelty of our study. On the contrary, the consistent observations in our study with the long-lasting conclusions supported the reliability of our analytical methods in collection and integration of heterogeneous datasets.

We respectfully disagree with the reviewer on the novelty concern. Building upon the reliable datasets of over 30,000 patients integrated from heterogeneous sources, we stressed our major contribution on the development of a novel computational method that:

- A. identified Metastasis-Featuring Primary (MFP) subtype (**Figure 2**), supported by clinical (**Figure 3d**) and independent molecular evidences (**Figure 3b and c**); Using machine learning, we achieved high accuracies in distinguishing metastases from primaries in breast and prostate cancers. More importantly, we defined MFP subtypes, a distinguishing group of primary patients with metastasis-enriched genetic variants. We then demonstrated the enormous clinical value of MFP in predicting patient prognosis.
- B. discovered and verified the high *MTOR*-inhibitor sensitivity in brain metastasis of lung cancer (**Figure 5d-f**); We discovered a significant enrichment of PI3K pathway aberration in the brain metastasis, rather than liver metastasis, of lung cancer, and verified using our pharmacogenomic database that targeting *MTOR*, the downstream effector of PI3K pathway, is high effective in treating lung cancer brain metastasis.
- C. stratified primary tumors into subtypes with propensities of organotropic metastases (**Figure 6**); We proposed a novel prognostic system to model metastatic cascade, which stratified primary prostate cancers into conventional, bone-metastasis-featuring, and

liver-metastasis-featuring primaries. Further analyses demonstrated the powerful performance of our model in improving clinical stratification of cancer patients.

- D. has open-access to the community which can use our models to explore organ-specific enrichment of particular variant, or to assess metastatic risk of one primary tumor given its genomic profile. The website, namely MetaNet (Metastatic Network model, accessible at <https://wanglab.shinyapps.io/metanet>), provides a user-friendly interface to explore organ-specific enrichment of particular variant, or to assess metastatic risk of one primary tumor given its genomic profile.

We believe this comprehensive study presented in this manuscript advances translational cancer research and will provide a very useful tool for evaluating patient metastasis.

Reflected in the revised manuscript:

To highlight the contributions of our study and to avoid the confusions, we have improved our introduction by pointing out our novel aspects.

“In this study, we aimed to develop a Metastatic Network model (MetaNet) to assess metastatic risk and potential destination organs through collecting and analyzing a total of 32,176 pan-cancer DNA-sequencing samples. Using this big data cohort, we identified genomic biomarkers associated with common and organotropic metastases, and validated their utility in metastatic risk assessment at early stage using a machine-learning model to sort out a distinguishing subtype, namely Metastasis-Featuring Primary, with shorter disease-free survival than Conventional Primary patients. From the biomarkers associated with brain metastasis of lung cancer, we discovered a significant enrichment of PI3K pathway aberration and verified using our pharmacogenomic database that targeting *MTOR* is high effective in treating lung cancer brain metastasis. Using the organotropic biomarkers, we established a novel computational model that stratifies patients of primary prostate cancer into subgroups with propensities of bone or liver metastases to inform organ-specific examinations in follow-ups. To facilitate the metastatic risk assessment and other organotropic biomarkers validation, we developed a web application of MetaNet which is available at <https://wanglab.shinyapps.io/metanet>.”

Reviewer #4 (Remarks to the Author): Expert in machine learning and cancer genomics

General comments:

“The authors have developed a computational framework (MetaNet) that integrates clinical and sequencing data from 32,176 primary and metastatic cancer cases, to assess metastatic risks of primary tumors. Using this framework, the authors identified Metastasis-Featuring Primary (MFP) tumors, a subset of primary tumors with genomic features enriched in metastasis, and demonstrated their high metastatic risks with significantly shorter disease-free survivals and higher migratory potential. The MSK-IMPACT and FoundationONE data were used in the training and testing procedure, while the data from TCGA were used for independent validation. It should be noted that the machine learning aspect of the work is not novel but the application to an outstanding problem could lead to novel findings. Furthermore, the authors should be commended on the extent of the analysis that they have performed.”

Author response:

We thank the reviewer for the summary of our work.

Major comment 1:

“The Introduction suffers from a poor overview and does not properly represent the contents of the paper. Introduction is sorely lacking references to others who have performed similar work before, and needs to address the significance of the novelty of this investigation (it could be quite significant, but is not conveyed as such). The introduction needs to spend much more time explaining what exactly this investigation is doing, and address significant background which is overlooked or assumed by the authors. The introduction is hampered by strong grammatical mistakes which cause excessive confusion to the reader.”

Author response:

We thank the reviewer for pointing out the problem of **Introduction** Section. In the original manuscript, we structured the **Introduction** into three parts: (1) Metastasis is associated with cancer deaths, but it is unpredictable using the current clinical, imaging and histological metrics. (2) We hypothesize that genomic variation of primary tumors could be used as the indicators for metastatic risk assessment. (3) Therefore, we aimed to develop a Metastatic Network model

(MetaNet) to assess metastatic risk and potential destination organs through collecting and analyzing a total of 32,176 pan-cancer DNA-sequencing samples.

Enlightened by the reviewer, we further cited two epidemiological studies (PMID: 32562887 and 32363344) to stress the importance of predicting potential metastatic destination, as it is highly related to patient survival. Moreover, we have added two representative references. One study (PMID: 12469122) used microarray data to predict metastasis and confirmed the predictive expression signature by the evidence of worse clinical outcome in the patients carrying this signature in their primary tumors, which is similar with our study, but we used genomic other than transcriptomic data. The other study (PMID: 25024180) used DNA sequencing data and extracted a predictive feature namely copy-number alteration burden to predict the relapse of prostate cancer. In our study, we used copy-number alteration burden as well with extra eight genomic features to achieve organotropic stratification of prostate cancer (**Figure 6**). And hence, we posit that the novelty of our study still holds.

Reflected in the revised manuscript:

“High throughput technologies enable identification of molecular signatures predictive of cancer metastasis and progression. In 2003, one of the seminal studies identified a gene expression signature associated with metastasis from microarray data, and found that the patients with this signature in their primary tumors were associated with metastasis and poor clinical outcome ⁴. Clinical tumor DNA sequencing methods, such as MSK-IMPACT ⁵ and FoundationONE ⁶, have demonstrated its clinical utility in guiding treatment selection in both primary and metastatic cancers ^{7,8}. Beyond treatment selection, copy-number alteration burden, an engineered genomic feature, proved to be highly predictive of the relapse of prostate cancer ⁹. We therefore hypothesize that genomic variation of primary tumors could be used as the indicators for metastatic risk assessment.”

“Epidemiological studies have discovered that depending on the tissue of origins and other factors, metastatic tumor cells have preference to seeding at certain distant organs, known as organotropism ^{11,12}. For example, the epidemiological data from 2010 to 2015 have shown that approximately 80% of synchronous brain metastases originated from lung primaries ¹³. Metastases from the same tissue but colonizing at different organs may result in different

survivals^{14,15}. In particular, hepatitis metastases, as the most prevalent diagnosis, lead to significantly worse clinical outcome than non-hepatic metastases in most cancer types¹⁶.”

“In this study, we aimed to develop a Metastatic Network model (MetaNet) to assess metastatic risk and potential destination organs through collecting and analyzing a total of 32,176 pan-cancer DNA-sequencing samples. Using this big data cohort, we identified genomic biomarkers associated with common and organotropic metastases and validated their utility in metastatic risk assessment at early stage using a machine-learning model to sort out a distinguishing subtype, namely Metastasis-Featuring Primary, with shorter disease-free survival than Conventional Primary patients. From the biomarkers associated with brain metastasis of lung cancer, we discovered a significant enrichment of PI3K pathway aberration and verified using our pharmacogenomic database that targeting *MTOR* is high effective in treating lung cancer brain metastasis. Using the organotropic biomarkers, we established a novel computational model that stratifies patients of primary prostate cancer into subgroups with propensities of bone or liver metastases to inform organ-specific examinations in follow-ups. To facilitate the metastatic risk assessment and other organotropic biomarkers validation, we developed a web application of MetaNet which is available at <https://wanglab.shinyapps.io/metanet>.”

Major comment 2:

“The authors hypothesize that genomic variations of primary tumors could be used as indicators for metastatic risk assessment. However, currently available genomic databases have only a small number of paired primary/metastatic samples, which are insufficient to sort out reliable prognostic biomarkers applicable in a larger cancer population. However, the authors have proposed to use unpaired sample data (derived from different patients) to characterize tumor aggressiveness. Using a computational pipeline, the authors train a classifier to differentiate between primary and metastasis and then try to link the findings to clinical outcome. In my opinion, the authors should definitely try to directly link the cases to outcome (as the final end-point) rather than using metastasis as an intermediate end-point. There are well-characterized algorithms that directly link features (genomics variations in this case) to outcome data (e.g., overall survival, disease free survival). Examples are cox-regression, cox-lasso, survival random forests, etc. Without this analysis, the reviewer believes that we cannot fairly assess the contribution of the proposed computational approach (i.e.,

using metastasis as the end-point and then link to survival versus directly linking genomic variation to survival).”

Author response:

We agree with the reviewer that compared to our method, directly using genomic information for predicting patient survival may achieve better result, because the direct method linked prognosis-related genes to stratify patients. However, the goal of our study was to stratify patients with primary tumors into various subtypes according to their metastatic risk and potential organotrophic properties, rather than to predict patient survival. Prediction of patient survival is not equivalent to risk assessment of tumor metastasis, because cancer patient survival relies on multiple complex factors including not only metastasis but also patient physiological condition and therapeutic solution. In addition to metastatic risk assessment, our model points out the organs at which metastatic cancer cells are most likely to colonize. Those organs hereby should be closely monitored during the follow-ups.

To achieve this goal, our study attempted to identify metastasis-related genes and then using these genes to assess the metastatic risk of patients (**Figure 2f**). To validate our metastatic risk assessment, we compared the disease-free survival (DFS) between the two groups and showed that the patients stratified into high metastatic-risk group by our model had significantly shorter DFS (**Figure 3d**). In contrast, directly using all the genes to predict survival of cancer patients has been intensively studied, and this method usually highlighted a set of survival-related genes including confounding factors irrelevant to metastasis. In conclusion, our analytical framework sorted out metastasis-related genes from the set of survival-related genes, which is indeed the major novelty of our study compared to previous studies on genomics-prognosis association. In addition, as mentioned in the Response to the **Major comment 1**, similar analytic methodology (PMID: 12469122) has been used to dissect metastasis-related genes using microarray data, which have proven to be predictive of patient survival.

Reflected in the revised manuscript:

To emphasis this point, we have added the following notes to the **Discussion** Section to distinguish our strategy from traditional studies by directly learning prognostic markers:

“Compared to the low-risk group, the high-risk group of patients have turned out to suffer significantly shorter disease-free survival with elevated migratory program significantly enriched in the transcriptome of their tumors. Different from previous studies that identified prognostic genomic biomarkers to predict patient survival ⁹, MetaNet focused on the biology of metastasis and identified 30 prevalent (fraction > 5%) and significant (FDR < 0.05) variants enriched in organotropic metastasis from a big-data perspective (Fig. 7a). This molecular portrait of organotropic metastasis exhibits strong potential to inform treatment selection (Fig 5d-f), and surveillance of drug resistance (Fig 5a-c) and distant metastasis (Fig 5g-i).”

Major comment 3:

“On Page 28, the authors state that “A threshold was selected to achieve the best separation of conventional primary group and metastasis-featuring primary group using disease-free survival time in TCGA.” This reviewer believes that the thresholds cannot be set based on the “unseen test data”. The authors need to select the thresholds based on the training and/or validation data and should NOT touch the test set (TCGA data) to achieve the best results. This is a major concern that makes it difficult to assess the validity of the findings.”

Author response:

There are two machine learning models in our study: one is to distinguish metastatic cancers from primary ones, and the other is to assess the metastatic risk of prostate cancer toward bone or liver. In the first model, we trained the model using the sequencing data from the three cohorts: MSK (PMID: 28481359), MET500 (PMID: 28783718) and FMI (PMID: 28235761), all of which did not provide the corresponding disease-free survival data in their publications. And hence, it is infeasible to “select the thresholds based on the training and/or validation data”.

In the original manuscript, we adopted a method of threshold selection from Uhlen *et al.* (*Science* 2017, PMID: 28818916) which “selected the value yielding the lowest log-rank P value”. To address the reviewer’s major concern and “*assess the validity of the findings*”, we conduct a comprehensive analysis on how the threshold selection affects the result of DFS validation using the four subtypes of the breast cancer as an example, as shown in the figure below. We show the probability density function (PDF) of the metascore computed by our model for the breast cancer patients in TCGA cohort at the first row of the figure. We then rank

the n patients by their metascores from the highest to the lowest, and enumerate all $n-1$ possible cutoff and calculate the log-rank test P-values (shown at $-\log_{10}$ scale), as shown in the second row of the figure. The black dots show the cutoff at the lowest P-values.

We next propose two additional methods to select the threshold: a quantile-based method and an unsupervised adaptive method. The quantile method selects the top k -quantile as the threshold, which was also adopted by other studies, e.g., Savas *et al.* (*Nature Medicine* 2018, PMID: 29942092) that selected the top 25% patients ranked by their expression signature. In our study, we select the top 10% patients ranked by our metascore as the thresholds (marked as black squares in the figure), and yield a significant DFS prognosis similar to the lowest-P method (marked as black dots).

To derive a fairer method of threshold selection, we propose the third method using Gaussian Mixture Models (GMM), an unsupervised method that can adaptively learn a threshold to separate data points into a predetermined number of clusters. We assume that there are two clusters of patients: one is conventional primary and the other is metastasis-featuring primary, and their metascores follow two different Gaussian distributions. Fitting the metascores using the GMM, we plot the PDF of the GMM onto the histograms at the first row of the figure (black curves), and the model adaptively derives a threshold to separate the patients from the two different Gaussian distributions (marked as black star signs). This unsupervised method also yields a significant DFS prognosis similar to the lowest-P method. Indeed, it apparently shows

at the second row of the figure that there exist multiple thresholds in the middle of the metascore-ranked patients, and hence we conclude that our metascore is robust in prioritizing patients with short disease-free survival.

The following description has been added to the **Methods** Section to elucidate the way in which we chose the thresholds:

“In order to determine a robust threshold of Metascore to separate the conventional primary group and the metastasis-featuring primary group, three different statistical methods were used:

(a) **Lowest-P**: Use the Metascore of each patient as the threshold and calculate the p value of log-rank test by comparing the disease-free survival between the two groups, and select the Metascore that yields the lowest p value as the threshold;

(b) **Top N%**: Rank all the patients based on the Metascore at descending order and select the top 10% of patients as the high-risk group; and

(c) **Unsupervised**: A Gaussian Mixture Model (GMM) with two components was used to fit the distribution of Metascore, and the threshold was at the intersection of the two components.

The final threshold was determined as the median of the candidate thresholds calculated by the three methods.”

In the second model, we utilized ordinal regression to model the metastasis of prostate cancer from primary site to bone (proximal dissemination) or to liver (distal dissemination). Through learning from the training data from the MSK and FMI cohorts, this model can quantify the metastatic risk of one sample by an organotropism score and two thresholds (0.1568 and 1.8058) to split the training samples into three classes:

- (1) Conventional Primary (CP) Group: $\text{organotropism_score} < 0.1568$,
- (2) Bone-Metastasis-Featuring Primary (Bone-MFP) Group: $0.1568 \leq \text{organotropism_score} < 1.8058$,
- (3) Liver-Metastasis-Featuring Primary (Liver-MFP) Group: $\text{organotropism_score} \geq 1.8058$.

Note that these two threshold values were automatically determined by the model from the training data and were absolutely independent of the TCGA data used in the validation. For more technical details about how the model learns the threshold as latent variables, please refer to the comprehensive survey on ordinal regression model by Gutiérrez *et al.* (KDD 2016). To

stress this self-adaptive thresholding of the model, we have added the following notes to the **Discussion** Section:

“In addition, unlike traditional multi-class models that consider the classes to be independent of each other, we proposed the ordinal regression with self-adaptive thresholding to model metastatic dissemination from tissues of origin to proximal sites and distant organs.”

Major comment 4:

“The paper lack details to reproduce the results. Given that “Machine learning” is incorporated in the title of the paper, I highly suggest that the authors make sure that they include enough details to enable others to reproduce results – this could be in the form of releasing code as well.”

Author response:

For our readers to reproduce our results, we provide the main codes used in this study at our GitHub repository: <https://github.com/WangLabHKUST/METANET-analysis>. Note that part of the datasets are under controlled access and cannot be deposited at this repository. For example, the processed data of FoundationONE are deposited at the dbGaP (Study Accession: phs001179.v1.p1). Please follow the instruction on the dbGaP website to request the data access (https://www.ncbi.nlm.nih.gov/projects/gap/cgi-bin/study.cgi?study_id=phs001179.v1.p1). For more details about the regulation of data usage and the approval we received, please see the **Comment 4** from the **Reviewer 1** and our response.

Conclusion:

“Taken together, while commending the extent of the work that has been performed, this reviewer believes that the manuscript requires substantial improvement to allow a fair assessment of the findings and contributions.”

Author response:

We have carefully addressed all the reviewers’ comments and revised the entire manuscript. And we wish the current revised version could receive “a fair assessment of the findings and contributions”.

REVIEWER COMMENTS

Reviewer #1 (Remarks to the Author):

The authors have addressed many of my concerns. I'm still a little confused about the grouping of cancers, however. I wonder if the authors could clarify their grouping of the cancers.

For example, "nasopharynx" is part of its own group and a group called "Head_and_Neck_excl_Nasopharynx". Chest wall is part of chest wall NOS. Bone is part of 4 categories: bone and joints, CNS, NOS, and soft tissue. What does this coding mean?

Also, for a category like "Head_and_Neck_excl_Nasopharynx", why just exclude nasopharynx? Are you excluding skin cancers as well? Radiation oncologists who treat head and neck cancer typically are also the ones who treat skin cancer (since this is where most of them form), and skin cancer is the number 1 cancer in most of the developed world, but metastases from it tend to be pretty rare.

Can the author provide some frequency tables of the number of cancers in each category, the histology in each, and the primary? Could they make some figures to go along with this? I feel like I am missing a big step somewhere before all of the impressive analyses in the figures.

Reviewer #2 (Remarks to the Author):

I'm satisfied with the revision.

Reviewer #3 (Remarks to the Author):

This is a revised manuscript describing the authors use of publicly available genomics data to try to build classifiers to predict metastatic progression. Although the authors have made some improvements to the manuscript, significant concerns about novelty remain. Comparisons to existing classifiers (ex. OncotypeDX, MammaPrint) are not comprehensively presented and many of the observations appear to be validation of existing knowledge. Importantly, since the genomic data is from patients who often were treated therapeutically, much of the interpretation of the results may be for acquisition of drug resistance, rather than propensity to metastasize. This is frequently a significant concern for this sort of analysis, since it is not possible to completely disentangle drug resistance and metastasis from overall survival data. While this is obviously important, the analysis would need to be re-focused to try to tease this aspect of the analysis out in a more comprehensive manner. As it stands, it is not clear to me that this manuscript represents a significant advance to warrant publication.

Reviewer #4 (Remarks to the Author):

The authors have addressed my comments.

Point-by-point Response* to Reviewers' Comments#

*Sentences excerpted from the manuscript are marked in blue

#All comments from reviewers are cited in quoted format

Response to Reviewer #1: Expert in metastasis and epidemiology

Comment 1:

"The authors have addressed many of my concerns. I'm still a little confused about the grouping of cancers, however. I wonder if the authors could clarify their grouping of the cancers. For example, "nasopharynx" is part of its own group and a group called "Head_and_Neck_excl_Nasopharynx". Chest wall is part of chest wall NOS. Bone is part of 4 categories: bone and joints, CNS, NOS, and soft tissue. What does this coding mean? Also, for a category like "Head_and_Neck_excl_Nasopharynx", why just exclude nasopharynx? Are you excluding skin cancers as well? Radiation oncologists who treat head and neck cancer typically are also the ones who treat skin cancer (since this is where most of them form), and skin cancer is the number 1 cancer in most of the developed world, but metastases from it tend to be pretty rare. Can the author provide some frequency tables of the number of cancers in each category, the histology in each, and the primary? Could they make some figures to go along with this? I feel like I am missing a big step somewhere before all of the impressive analyses in the figures."

Author response:

We agree with the reviewer that the way in which we presented the cancer grouping process is not clear enough. In this revision, we clarify the logic behind and provide corresponding supportive evidence. First, we re-format the **Supplementary Table 1** and **2** in order to provide "the number of cancers in each category", and the diagnostic details in "histology" curated from the original curated datasets. Second, we visualize the process going along with the cancer grouping in **Supplementary Figure 1b**. And third, we perform cancer grouping by integrating evidence from (1) the OncoTree (<http://oncotree.mskcc.org/>), an online software developed by Memorial Sloan Kettering Cancer Center (Kundra *et al.* JCO 2021), (2) the WHO Classification system on pathologyonline (<https://www.pathologyoutlines.com/books?pub=7>), and (3) the consensus of our pathologist panel. Using this method, we categorize the cancer samples from 207 tissue sites into 41 general primary sites, and only 192 out of 29,450 (0.65%) samples give

rise to discrepancy between OncoTree and our pathologists. Most of these samples are located at bones or soft tissues with ambiguous cancer types, and we did not enroll them in our study to ensure the solidity of our analysis. The revised cancer grouping result, together with our clarification notes when the discrepancy take places, is presented in the updated **Supplementary Table 1** and **2**. We show part of grouping results from **Supplementary Table 1** below and clarify the meaning of each column.

Original Primary Tumor Site	Number of Samples	Common Diagnosis	OncoTree Annotation	Merged by Pathologists	Notes	Is Enrolled	Tissue Name Shown in Figures
Breast	4,026	Carcinoma, NOS // Infiltrating duct carcinoma, NOS // Breast Invasive Ductal Carcinoma	Breast	Breast	/	enrolled	Breast
Bronchus And Lung	3,769	Adenocarcinoma, NOS // Non-small cell carcinoma // Squamous cell carcinoma, NOS	Lung	Lung	/	enrolled	Lung
Colon	1,858	Adenocarcinoma, NOS // Mucinous carcinoma // High-Grade Neuroendocrine Carcinoma of the Colon and Rectum	Bowel	Colorectal	Bowel includes small intestine, colon, rectum and anus	enrolled	Colon

For the concrete concerns of the reviewer, we clarify as below. First, we consider nasopharyngeal carcinoma (NPC) and other head and neck cancers to be separate groups as they are pathologically, epidemiologically and genomically different. Unlike part of other head and neck cancers caused by human papillomavirus (HPV), NPC is primarily caused by the infection of Epstein-Barr virus (EBV) with featuring regional epidemiology in South China. In addition, over a half of HPV-negative head and neck cancers collected by TCGA showed enriched *TP53* mutation and chromosomal instability (PMID: 25631445), while EBV-negative NPCs are rare and the NPC genomes are stable (PMID: 24952746). By ruling out NPC from other head and neck cancers, we found that the brain metastasis of head and neck cancers exhibits higher chromosomal instability than the primary head and neck cancers (**Supplementary Figure 4d**). Therefore, we classify NPC as a separate group from other head and neck cancer.

Second, for those cancers located at ambiguous places such as chest wall, bone, and soft tissue, we explain how we define the classification in the **Notes** column of **Supplementary Table 1**.

Most of those cancers do not belong to one pathologically unique class, and hence we did not enroll them in our study, which ensures that the cancer classification prior to our analysis does not impair the results of our entire study.

Third, in this revision, we “*make some figures to go along with this*” process of cancer grouping in **Supplementary Figure 1b**, also shown below, to visualize how we performed sample filtering and selection in details.

Reflected in the revised manuscript:

We have updated **Supplementary Table 1** and **2** with the revised grouping of cancer types, and added **Supplementary Figure 1b** and corresponding figure explanation for visualization of cancer grouping workflow. In particular, we have added the following in the **Methods** Section.

“For convenience of downstream study, we merged the tissues into 47 anatomical organs (Supplementary Table 1 and 2) via a computational cancer classification system, OncoTree⁶⁵, followed by the consensus of a pathologist panel (Supplementary Fig. 1b).”

Reviewer #2 (Remarks to the Author): Expert in machine learning and genomics

General comment:

“I'm satisfied with the revision.”

Author response:

We thank the reviewer for the helpful comments that dramatically improved our study.

Reviewer #3 (Remarks to the Author): Expert in metastasis and cancer genomics

Comment 1:

“This is a revised manuscript describing the authors use of publicly available genomics data to try to build classifiers to predict metastatic progression. Although the authors have made some improvements to the manuscript, significant concerns about novelty remain. Comparisons to existing classifiers (ex. OncotypeDX, MammaPrint) are not comprehensively presented and many of the observations appear to be validation of existing knowledge.”

Author response:

We would like to pinpoint the methodological difference, superior performance, and unique novelty of our MetaNet in comparison with OncotypeDX and MammaPrint.

First, the strategies used in the development of our MetaNet and the two existing classifiers are methodologically different. OncotypeDX is a commercial product that uses an RT-PCR (Reverse-Transcriptase-Polymerase-Chain-Reaction) assay of 21 genes to predict the likelihood of distant recurrence in breast cancer patients (Paik *et al.* NEJM 2004). Similarly, MammaPrint is another commercial product that uses a microarray assay of 70 genes to predict clinical outcome of breast cancer patients (van 't Veer *et al.* Nature 2002). Both classifiers share three key strategies in the development: (1) they both use the expression data of a small gene set; (2) the machine-learning models used in the two classifiers are linear methods, assuming that the molecular features are independent; and (3) they both validated the clinical utility of

their methods mainly in ER+ (Estrogen Receptor positive) breast cancer patients. In contrast, our MetaNet (1) relied on DNA somatic variants rather than expression profile; (2) utilized xgboost, a non-linear classification method that can capture complex dependencies between molecular features; and (3) demonstrated the performance of prognosis prediction in all four subtypes of breast cancer (**Figure 3e** and **Figure S3f**).

Second, we have followed the reviewer's comment and compared our MetaNet with OncotypeDX to predict disease-free survival of the breast cancer patients in TCGA cohort. Following the limited disclosed description regarding the OncotypeDX methodology (Paik *et al.* NEJM 2004), we sorted out the 21 genes used in OncotypeDX from the RNA-seq data of TCGA breast cancer cohort, and re-implemented the model to score the recurrence likelihood of each patient. Using the same cutoff with the one provided by the original paper (Paik *et al.* NEJM 2004), we stratify the patients into three groups: the patients with high, intermediate, and low risks of recurrence. To validate the prediction, we show the disease-free survival of the three stratified groups in the following figure (left panel), and find that in TCGA breast cancer cohort, the stratification of OncotypeDX is not as good as the one by our MetaNet shown in our manuscript (**Figure 3d**, also shown in right panel).

Third, even though several studies have demonstrated tumor molecular features are predictive of distance recurrence in the past two decades, we stress that our study, for the first time, proposed to predict organ-specific metastasis using tumor genomic variants. Serving as one of the important novelties in our study, the risk stratification of organotropic metastasis (**Figure**

6) can inform clinicians of organ-specific examination in follow-ups, which cannot be achieved by any existing classifiers.

Taken all the above evidence together, we respectfully disagree the reviewer on the comment regarding the novelty of our study. While we believe that our MetaNet has better performance of cancer patient stratification in disease-free survival of TCGA cohorts, we acknowledge that in most cases, direct comparison was not performed in large-scale clinical practice. And we believe a better method might emerge when combining expression with somatic variants using an ensemble model with linear and non-linear classifiers trained by a large-scale independent cohort.

Reflected in the revised manuscript:

To stress the advantages of our methods in comparison with the existing classifiers, we have added the following sentences in the **Discussion** Section.

“Many previous studies have proven that tumor molecular features are highly predictive of disease progression and drug response of cancer patients. One longstanding strategy is to develop machine-learning models that learn the likelihood of tumor recurrence or metastasis from a small set of signature genes ⁴. This strategy has been commercialized into widely used diagnostic products in breast cancer, such as OncotypeDX ⁶⁰ and MammaPrint ⁶¹. Unlike this strategy, the novelty of MetaNet lies in the use of somatic variants from a large-scale pan-cancer cohort including 32,176 primary and metastatic samples, the development of machine-learning models using a non-linear classifier with highly informative interpretability, and the new application in risk stratification of organ-specific metastases. While direct comparison has not been performed between the two methodologies, we believe that a better method might emerge through integrating genomic and transcriptomic data, or even data from digital pathology using complex classifiers like deep learning.”

Comment 2:

“Importantly, since the genomic data is from patients who often were treated therapeutically, much of the interpretation of the results may be for acquisition of drug resistance, rather than propensity to metastasize. This is frequently a significant concern for this sort of analysis, since it is not possible to completely disentangle drug resistance and metastasis from overall survival

data. While this is obviously important, the analysis would need to be re-focused to try to tease this aspect of the analysis out in a more comprehensive manner. As it stands, it is not clear to me that this manuscript represents a significant advance to warrant publication.”

Author response:

We agree with the reviewer that for the patients who develop metastasis after treatment, it is challenging to distinguish the genomic variants associated with drug resistance or metastasis. However, we argue that the variants associated with organ-specific metastases identified by our study (**Figure 4, 5 and 6**) are highly likely associated with metastasis rather than drug resistance, since rare study has proven that drug resistance can give rise to organ-specific metastases. In contrast, many previous studies have demonstrated that metastatic organotropism is associated with tissue of origin, blood vasculature, and congenial microenvironment of distant organs (PMID: 31063756, 28741564, 20610625). In our manuscript, we developed MetaNet to chart a comprehensive map of genomic variants associated with organotropic metastasis using an integrated cohort of 30,000+ samples. As a result, we derived a list of 30 prevalent variants enriched in organotropic metastasis, as shown in **Figure 7a** and Supplementary Table 3. Except *ESR1* mutation in breast cancer that has proven to be acquired against endocrine therapy (PMID:25928204), most of the variants in the list are shown to connect to cancer development, such as *TP53* mutation, *KRAS* mutation, and *MYC* amplification. We summarize the literature evidence regarding the role of these variants in metastasis in the following table.

Primary Cancer Type	Variant Label	Site with Max Mut Fraction	Literature Evidence (PMID)
Breast	CDH1_mut	Bone	Dominant in invasive lobular carcinoma, related to discohesive morphology related to tumor invasion and metastasis (26451490)
Breast	CDK12_amp	Brain	Proximal to ERBB2
Breast	DDR2_amp	Brain	Related to metastasis (31144616)
Breast	ERBB2_amp	Brain	Well-known subtype biomarker, enriched in brain metastasis of breast cancer (17541441)
Breast	ESR1_mut	Liver	Acquired resistance under endocrine therapy, enriched in liver metastasis (25928204)
Breast	MYC_amp	Brain	Often acquired in metastasis (22056952)
Breast	MYC_pathway	Brain	Regulated by MYC_amp
Breast	p53_pathway	Brain	Regulated by TP53_mut
Breast	RAS_pathway	Brain	Regulated by ERBB2_amp

Breast	TP53_mut	Brain	Enriched in brain metastasis of breast cancer (22187033)
Colon	BRCA2_amp	Brain	Proximal to CDK8
Colon	CDK8_amp	Brain	Related to colorectal cancer initialization (18794900), highly likely related to metastasis (this study)
Colon	chr13q_amp	Brain	Encompass CDK8
Colon	FLT1_amp	Brain	Proximal to CDK8
Colon	FLT3_amp	Brain	Proximal to CDK8
Colon	KRAS_mut	Brain	Innate resistance under EGFR-targeted therapy (17375050)
Colon	RAS_pathway	Brain	Regulated by KRAS_mut
Colon	RB1_amp	Brain	Proximal to CDK8
Colon	TGF_beta_pathway	Liver	Featured by SMAD4 loss, enriched in distant metastasis (10340381), promote metastasis (19909744) and drug resistance (16144935)
Lung	CREBBP_mut	Brain	Enriched in small-cell lung cancer (27168435) and increase Pracinostat sensitivity (30181244)
Lung	EPHA5_mut	Brain	Unclear role in lung cancer
Lung	KRAS_mut	Bone	Innate resistance under EGFR-targeted therapy (19349489)
Lung	Notch_pathway	Brain	Regulated by CREBBP_mut
Lung	PI3K_pathway	Brain	Regulated by STK11_mut, enriched in brain metastasis (25929848), sensitive to PI3K inhibitor (this study)
Lung	RAS_pathway	Brain	Regulated by KRAS_mut
Lung	STK11_mut	Bone	A.k.a LKB1, related to initiation, differentiation and metastasis of lung cancer (17676035)

Furthermore, we think that metastasis-related variants need not to be mutually exclusive to drug resistor. Drug resisters can be divided into two main categories: acquired resistor and innate resistor. As mentioned above, *ESR1* mutation is a well-known acquired resistor that is often acquired under endocrine therapy and rarely found in primary treatment-naïve patients (PMID:25928204). In contrast, innate resisters such as *KRAS* mutation, have been present in many cancers before treatment and often serve as a negative predictor of drug response such as *EGFR* inhibitor. These innate resisters might be able to play a dual role not only in drug resistance but also metastasis, as tumor cells must be able to survive under treatment pressure before migrating to a distant organ. For example, previous studies have shown that *TP53* alterations are frequently connected to chemoresistance in many cancers (PMID:27888811), and get involved in many biological functions related to metastasis, such as cell motility and

invasion (PMID:31952783). And hence, drug resistance could serve as the first step toward a distant organ during the journey of metastasis. In this case, the dual role of a variant in drug resistance and metastasis cannot be taken apart.

In addition, although some drug resisters might be in the list of our discovered metastatic variants, our MetaNet has captured sufficient metastatic features and remained capable of predicting the risk of recurrence for treatment-naïve patients. To verify this hypothesis, we first scrutinize the clinical information of 1,101 patients within TCGA breast cancer cohort (downloaded from cBioPortal), and find that 98.27% of the patients did not receive any neoadjuvant therapy prior to resection. We then perform the same prediction as the one shown in **Figure 3** using MetaNet, and find that the predictability of MetaNet still holds in the treatment-naïve patients (see the Figure below, left panel). This result suggests that minor unknown signal from drug resisters does not affect the clinical utility of MetaNet in prognosis.

Finally, we acknowledge that we cannot rule out the potential drug resisters within our list of metastasis-associated variants, as it requires strict experiments to consolidate the biological roles of the discovered variants. In this revision, we stress this point in the **Discussion** Section to avoid misleading of our readers.

Reflected in the revised manuscript:

“Another limitation of our study is that we cannot rule out drug resistors from the list of metastasis-related variants based on the present statistical comparison. A better design might be the collection of treatment-naïve samples from the patients with synchronous metastasis. In this case, the statistical comparison of genomic profiles between primary and synchronous metastatic samples could unveil driver variants with high potential in facilitating metastasis independent of treatment solutions. Further validation through biological experiments is also a must to consolidate the biological roles of the discovered variants.”

Reviewer #4 (Remarks to the Author): Expert in machine learning and cancer genomics

General comment:

“The authors have addressed my comments.”

Author response:

We thank the reviewer for the helpful comments that dramatically improved our study.

REVIEWERS' COMMENTS

Reviewer #1 (Remarks to the Author):

My concerns have been addressed.

Reviewer #3 (Remarks to the Author):

Despite the authors attempts to address my concerns, I remain skeptical that this represents a fundamental advance for tumor prognosis, superior to existing expression based assays. Moreover, the concerns regarding therapy resistance have not been addressed. Neoadjuvant therapy resistance is not the concern for distant metastasis so much as resistance to adjuvant therapy, which was not addressed.

Response to Reviewers' Remaining Concerns

Reviewer #3 Comment:

“Despite the authors attempts to address my concerns, I remain skeptical that this represents a fundamental advance for tumor prognosis, superior to existing expression based assays. Moreover, the concerns regarding therapy resistance have not been addressed. Neoadjuvant therapy resistance is not the concern for distant metastasis so much as resistance to adjuvant therapy, which was not addressed.”

Author response:

We thank the reviewer for raising the remaining concerns. To address these concerns, we have

(1) further revised the discussion paragraphs to clarify that compared to the expression-based assays, our genomics based method estimates metastatic risk from a different perspective: *“While direct comparison has not been performed between the two methodologies in a large-scale dataset with genomic and transcriptomic information available, we believe that MetaNet, a genomics-based method, can provide a different perspective complementary to the expression-based assays. And it is anticipatable that a better method might emerge through integrating genomic and transcriptomic data, or even data from digital pathology using complex classifiers like deep learning.”*

(2) added the following sentences in the Discussion Section to clarify the limitations and caveats of our method in terms of drug resistance: *“Another limitation of our study is that we cannot rule out drug resistors from the list of metastasis-related variants based on the present statistical comparison. A better design to overcome this limitation could be the enrollment of patients without neoadjuvant or adjuvant chemotherapy. In this case, the statistical comparison of genomic profiles between primary and metastatic samples could unveil metastasis drivers independent of treatment solutions. Further validation through biological experiments is also a must to consolidate the biological roles of the discovered variants.”*